# Redundancy and the role of protein copy numbers in the cell polarization machinery of budding yeast

Fridtjof Brauns [1,2], Leila Iñigo de la Cruz[3], Werner K.-G. Daalman[3], Ilse de Bruin [3], Jacob Halatek[1], Liedewij Laan[3] ✉ & Erwin Frey [1,4] ✉

How can a self-organized cellular function evolve, adapt to perturbations, and acquire new sub-functions? To make progress in answering these basic questions of evolutionary cell biology, we analyze, as a concrete example, the cell polarity machinery of *Saccharomyces cerevisiae*. This cellular module exhibits an intriguing resilience: it remains operational under genetic perturbations and recovers quickly and reproducibly from the deletion of one of its key components. Using a combination of modeling, conceptual theory, and experiments, we propose that multiple, redundant self-organization mechanisms coexist within the protein network underlying cell polarization and are responsible for the module's resilience and adaptability. Based on our mechanistic understanding of polarity establishment, we hypothesize that scaffold proteins, by introducing new connections in the existing network, can increase the redundancy of mechanisms and thus increase the evolvability of other network components. Moreover, our work gives a perspective on how a complex, redundant cellular module might have evolved from a more rudimental ancestral form.

Biological systems are self-organized. Their function emerges by the collective interplay of many components—governed by physical and chemical processes. How do such collective (self-organized) functions evolve and adapt to strong perturbations such as the loss of essential components[1,2]?

A striking example for such adaptation is the Cdc42 cell-polarization machinery of *S. cerevisiae* (budding yeast). Cell polarization directs cell division of budding yeast through the formation of a polar zone with high Cdc42 concentration on the membrane (see Fig. 1a–c). Following the knock-out of Bem1, a key player in the Cdc42-interaction network (Fig. 1d), cells regain their ability to polarize and divide by loss of another component of this network. This happens rapidly (within 100 generations) and reproducibly[3]. How this recovery works has remained unclear.

Cell polarization of budding yeast is organized by a complex interaction network (Fig. 1d) around the central polarity protein Cdc42. Cdc42 is a GTPase that cycles between an active (GTP-bound) and an inactive (GDP-bound) state. The key features of these two states are that active Cdc42 is strongly membrane bound and recruits many downstream factors, while inactive Cdc42-GDP can detach from the membrane to the cytosol where it diffuses freely.

In wild-type (WT) cells, polarization is directed by upstream cues like the former bud-scar[4–7]. Importantly however, Cdc42 can polarize *spontaneously* in a random direction in the absence of such cues[8–10]. What are the elementary processes underlying spontaneous Cdc42 polarization? On the timescale of polarity establishment, the total protein copy number of Cdc42 proteins (as well as its interaction partners) is nearly constant. Hence, to establish a spatial pattern in the

[1]Arnold Sommerfeld Center for Theoretical Physics and Center for NanoScience, Department of Physics, Ludwig-Maximilians-Universität München, Munich, Germany. [2]Kavli Institute for Theoretical Physics, University of California Santa Barbara, Santa Barbara, CA 93106, USA. [3]Department of Bionanoscience, Kavli Institute of Nanoscience Delft, Delft University of Technology, Delft, the Netherlands. [4]Max Planck School Matter to Life, Hofgartenstraße 8, D-80539 Munich, Germany. ✉e-mail: l.laan@tudelft.nl; frey@lmu.de

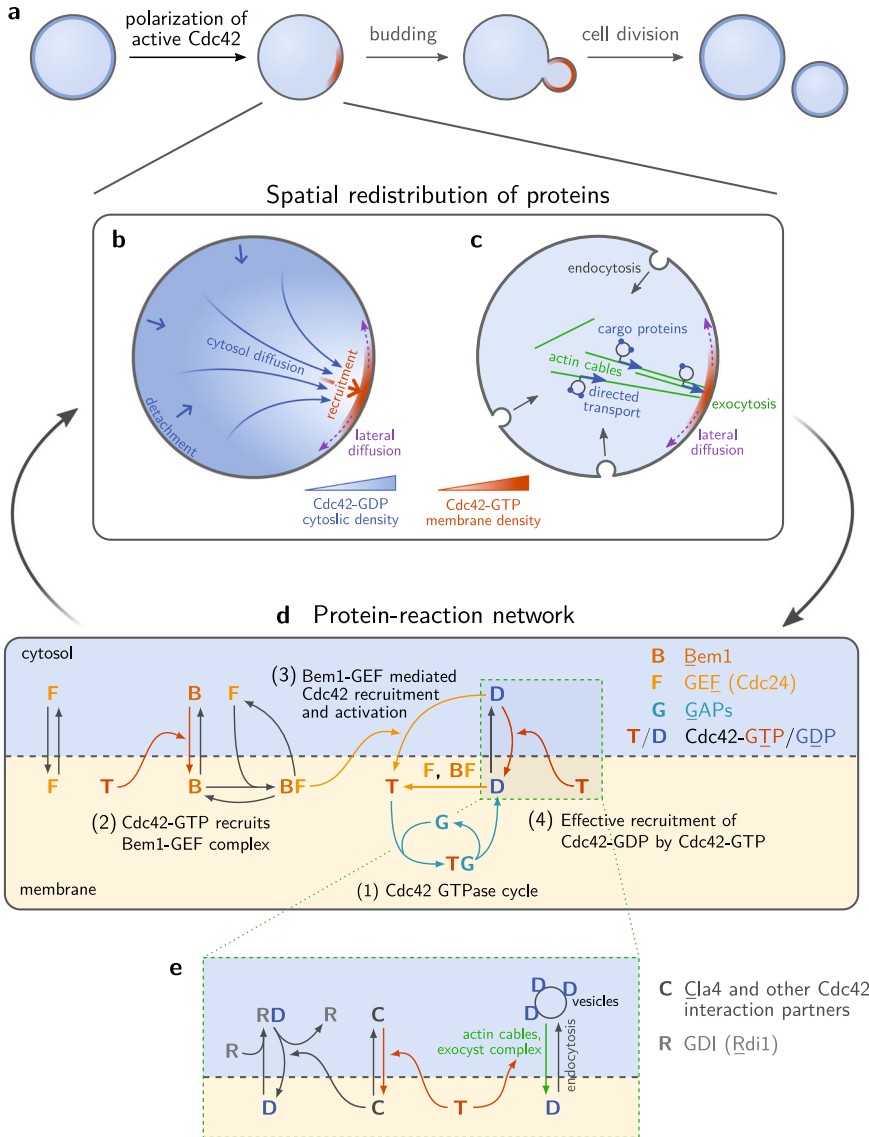

**Fig. 1 | Cell division of *S. cerevisiae* is spatially controlled by self-organized polarization of Cdc42. a** Starting from an initially homogenous distribution of Cdc42, a polar zone forms, marked by a high concentration of active Cdc42 on the plasma membrane. There are two pathways of directed transport in the cells: **b** Cytosolic diffusive flux driven by a concentration gradient that is sustained by spatially separated attachment (red arrow) and detachment (blue arrow) zones; **c** Vesicle transport (endocytic recycling) is directed along polar-oriented actin cables. Active Cdc42 directs both cytosolic diffusion (by recruiting downstream effectors that in turn recruit Cdc42) as well as vesicle transport (by recruiting Bni1 which initiates actin polymerization). **d** Molecular interaction network around the

GTPase Cdc42, involving activity regulators (GEF, GAPs), and the scaffold protein Bem1 (some components are displayed multiple times for visual clarity, not to imply a chronological order). An effective recruitment term accounts for Cdc42-recruitment to the membrane directed by Cdc42-GTP facilitated by Cdc42's interaction partners, for instance Cla4[13,31,73] and Rsr1[86] (**e**). Details of the model and the mathematical implementation are described in the Methods and Supplementary Note 1. For simplicity, we do not explicitly account for Cdc42-effector complexes. A model extension accounting for those complexes did not significantly change the results.

protein concentration, the so-called polar zone, the proteins need to be spatially redistributed in the cell by *directed transport*. There are two distinct, mostly independent, pathways for directed transport that have been established by experimental and theoretical studies[9–13]: cytosolic diffusive flux driven by a sustained concentration gradient (Fick's law) and vesicle-based active transport along polarized actin cables (Fig. 1b, c).

Once a polar zone has been established, the ensuing concentration gradient on the membrane leads to a diffusive flux of proteins away from the polar zone. To maintain the polar zone, this flux on the membrane must be counteracted continually by (re-)cycling the proteins back to the polar zone via a flux from the cytosol to the membrane[10,12,14] or via vesicle-based transport[9,11]. In WT cells,

Cdc42-GTP recruits Bem1 from the cytosol which in turn recruits the GEF (Guanine nucleotide Exchange Factor) Cdc24 (see Fig. 1d)[8,15]. The membrane-bound Bem1-Cdc24 complex then recruits more Cdc42-GDP from the cytosol and activates it (nucleotide exchange)[16]. The hallmark and crucial element of this *mutual recruitment mechanism* is the co-localization of Cdc42 and its GEF Cdc24[10,13,16–18].

Deletion of Bem1 disrupts localized Cdc42 recruitment and activation[13,18] and thereby severely impedes the cells' ability to polarize and bud[8,19]. *Bem1Δ* cells can be rescued by Bem1 fragments that cannot mediate mutual recruitment of Cdc42 and its GEF Cdc24, but only confer increased global (homogeneous) GEF activity by relieving Cdc24's auto-inhibition[20–23]. Even more intriguingly, in experimental evolution, *bem1Δ* mutants are reproducibly rescued by the subsequent

loss of Bem3[3], one of four known Cdc42-GAPs that catalyze the GTP-hydrolysis, i.e., switch Cdc42 into its inactive, GDP-bound state. These experimental findings suggests that there is a hidden Cdc42 polarization mechanism that is independent of GEF co-localization and is activated by either increased GEF activity or the loss of a Cdc42-GAP.

Here, we develop a mathematical model for the cell polarization module of budding yeast – synthesizing the insights from a large body of experimental and theoretical literature. Our theoretical analysis of this model shows that the cell-polarity module comprises multiple redundant mechanisms based on reaction–diffusion and potentially vesicle-based transport. It reveals that in addition to the Bem1-mediated mutual recruitment mechanism, a distinct and latent mechanism exists in the Cdc42-polarization machinery. Crucially, this latent mechanism requires explicit modeling of the intermediate Cdc42-GAP complex, which was not accounted for by previous models. We show that the latent mechanism operates under different constraints on the protein copy numbers than the wild-type mechanism and is activated by the loss of Bem3 which lowers the total protein copy number of GAPs. This explains how cell polarization is rescued in *bem1Δ bem3Δ* cells[3], and also reconciles the puzzling experimental findings outlined above. Moreover, we experimentally confirm the predictions of our theory on how cell polarization in various mutants can be rescued by changing the Cdc42 protein copy number. On the basis of the mechanistic understanding of the cell polarization module in budding yeast, we then propose a possible evolutionary scenario for the emergence of this self-organized cellular function. We formulate a concrete hypothesis whereby evolution might leverage scaffold proteins to introduce new connections in an existing network, and thus increase redundancy of mechanisms within a functional cellular module. This redundancy loosens the constraints on the module and thereby enables further evolution of its components, for instance by duplication and sub-functionalization[24].

## Results

As basis for our theoretical analysis, we first need to formulate a mathematical model of the cells' Cdc42-polarization machinery that is able to explain Bem1-independent polarization. The interplay of spatial transport processes (Fig. 1a, c) and protein-protein interactions (Fig. 1d) is described in the framework of reaction–diffusion dynamics. The biochemical interaction network we propose is based on the quantitative model introduced in[12] and makes several minimal, but essential extensions to it. The model accounts for the Cdc42 GTPase cycle and the interactions between Cdc42, Bem1 and Cdc24[10]. Extending previous models, we explicitly incorporate the transient formation of a GAP-Cdc42 complex as an intermediate step in the enzymatic interaction between GAPs and Cdc42[25]. Explicitly accounting for the enzyme kinetics of GAPs, which was neglected in previous models[26–28], is important to account for (partial) GAP saturation in regions of high Cdc42 concentration. Indeed, this enzyme saturation is a generic property of enzymatic kinetics. It will play an essential role in our findings. We also include effective self-recruitment of Cdc42-GDP to the membrane which is facilitated by membrane-bound Cdc42-GTP. This effective recruitment accounts for vesicle-based Cdc42 transport along actin cables[11,29,30] and putative recruitment pathways mediated by Cdc42-GTP downstream effectors such as Cla4 and Gic1/2[31–33]. A detailed description of the model, illustrated in Fig. 1d, and an in-depth biological motivation for the underlying assumptions are given in the Supplementary Note 1.

## The Cdc42 interaction network facilitates a latent polarization-mechanism

We first ask whether the proposed reaction–diffusion model of the Cdc42 polarization machinery can explain spontaneous polarization in the absence of Bem1, i.e. without GEF co-localization with Cdc42. To this end, we perform a linear stability analysis of the model which identifies the regimes of self-organized pattern formation. A large-scale parameter study (see Supplementary Note 4) reveals that in the absence of Bem1 there is a range of protein numbers of Cdc42 and GAP where polar patterns are possible (Fig. 2b), i.e., that there is a latent polarization mechanism. However, in contrast to the Bem1-dependent mutual recruitment mechanism (Fig. 2a), we find that the regime of operation for this latent mechanism is more limited and requires a sufficiently low GAP/Cdc42-concentration ratio (Fig. 2b). To validate the results from linear stability analysis, we performed numerical

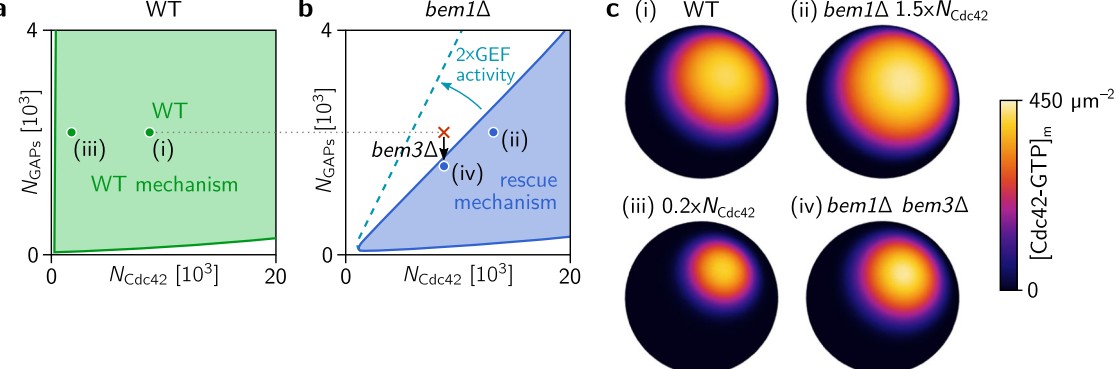

**Fig. 2 | Regimes of operation of the Bem1-mediated wild-type mechanism and the latent mechanism for cell polarity.** Stability diagrams as a function of GAP- and Cdc42 concentrations in presence and absence of Bem1 obtained by linear stability analysis (see Supplementary Note 2) of the mathematical model for the Cdc42-polarization machinery (see Methods). Shaded areas indicate regimes of lateral instability, i.e. where a spontaneous polarization is possible. **a** In WT cells, the scaffold protein Bem1 is present and facilitates spontaneous polarization by a mutual recruitment mechanism that is operational in a large range of Cdc42 and GAP concentrations[10,12]. The green point marks the Cdc42 and GAP concentrations of WT cells. **b** In the absence of Bem1, spontaneous polarization is restricted to a much smaller parameter-space region in our model, because the regime of operation of the Bem1-independenent mechanism is inherently is delimited by a critical ratio of GAP concentration to Cdc42 concentration. The Cdc42 and GAP

concentrations of *bem1Δ* cells and *bem1Δ bem3Δ* are marked by the red cross and blue point, respectively. The experimental observation that *bem1Δ* cells do not polarize, whereas *bem1Δ bem3Δ* polarize can be used to infer a range for the critical GAP/Cdc42-concentration ratio. Increasing the GEF activity of Cdc24 increases this critical ratio (dashed blue line). **c** Snapshots from numerical simulations showing the concentration of membrane bound Cdc42-GTP in the final steady state for various mutant and copy number conditions, corresponding to Videos 1–4. (In panel (iii), the color bar represents concentrations in the range 0–200 μm⁻².) (Model parameters were obtained by sampling for parameter sets that are consistent with the experimental findings on various mutants, as described in detail in Supplementary Note 4; see Supplementary Figs. S3 and S4 and Supplementary Tables S2–S4).

simulations of the full nonlinear, bulk-surface coupled reaction diffusions (see Fig. 2c and Videos 1–6; details described in Supplementary Note 3).

What is the mechanistic cause for the constraint on the GAP/Cdc42-concentration ratio? To answer this question, we need to understand how the Cdc42-polarization mechanism works in the absence of Bem1. As emphasized above, Cdc42-polarization requires two essential features—directed transport of Cdc42 to the polar zone and localized activation of Cdc42 there. The first feature, directed transport, is accounted for in the model by effective recruitment of Cdc42-GDP to the membrane mediated by active Cdc42 (Fig. 1d).

### GAP saturation can localize Cdc42 activity to the polar zone

How is the second feature, localization of Cdc42 activity to the polar zone, implemented in the absence of Bem1? Instead of directly increasing the rate of Cdc42 activation in the polar zone (via recruitment of the GEF Cdc24 by Bem1), localization of activity can also be achieved by decreasing the rate of Cdc42 *deactivation* in the polar zone and increasing it away from the polar zone. In fact, if enzyme saturation limits the net deactivation rate, a simple increase in Cdc42 density *generically* leads to a decrease of the Cdc42 deactivation rate (per Cdc42 molecule). Enzyme saturation of catalytic reactions occurs when the dissociation of the transient enzyme-substrate complex (here the GAP-Cdc42 complex) is the rate limiting step. The enzymes that are transiently sequestered in enzyme-substrate complexes are then not available to bind to further substrate molecules. Indeed, it has been shown that this is the case for GAP-catalyzed hydrolysis of Cdc42 in budding yeast[25]. Furthermore, enzyme saturation requires that a large fraction of enzymes is sequestered in enzyme–substrate complexes, i.e., that the total enzyme density is sufficiently low compared to the substrate density, as we found in the linear stability analysis (Fig. 2b).

In summary, (partial) GAP saturation localizes Cdc42 activity to the polar zone: It decreases the deactivation rate in the polar zone, where Cdc42 density is high, relative to the remainder of the membrane, where Cdc42 density is low. This localized Cdc42 activity, in conjunction with transport of Cdc42 to the polar zone, drives spontaneous cell polarization. Interestingly, enzyme saturation of Cdc42 hydrolysis is one of the six theoretically possible mechanisms for pattern formation that were hypothesized by a generic mathematical analysis of feedback loops in GTPase cycles[34].

### The latent polarization-mechanism explains the rescue of Bem1 deletion

The Bem1-independent rescue mechanism requires a sufficiently low GAP/Cdc42-concentration ratio to be functional (Fig. 2b). This suggests that *bem1Δ* cells are not able to polarize because their GAP protein copy number is too high. Our model predicts that the loss of GAPs can rescue cell polarization by bringing their total protein copy number into a regime where the Bem1-independent mechanism is operational, as indicated by the arrow in Fig. 2b. This is in accordance with evolution experiments showing that *bem1Δ* cells are reproducibly rescued by a subsequent loss-of-function mutation of the GAP Bem3[3]. Bem3 accounts for approximately 25% of the total protein copy number of all Cdc42-GAPs[35], indicating that *bem1Δ* mutants are close to the GAP/Cdc42-ratio threshold of the Bem1-independent mechanism. This proximity of the protein copy numbers to the threshold explains why a low fraction (about 1 in $10^5$) of mutants are able to polarize and divide, after *BEM1* has been deleted[3]: Protein copy numbers vary stochastically from cell to cell such that a small fraction of cells lies in the concentration regime where the latent polarization mechanism drives spontaneous cell polarization. (For the four Cdc42 GAPs, a coefficient of variation around 0.14 for cell-to-cell copy-number variability has been reported[36]. This is on the same order of magnitude as the upper estimate of 25% for the GAP protein copy number reduction required

to activate the Bem1-independent rescue mechanism, suggesting that this mechanism is operational in a fraction of *bem1Δ* cells.)

Rather than by the loss of a GAP, the GAP/Cdc42-concentration ratio could also be brought down by an increase of the Cdc42 protein copy number. Yet another option would be an increase of Cdc24's GEF activity which would increase the critical threshold in GAP/Cdc42-concentration ratio (see dashed line in Fig. 2b). However, compared to a loss-of-function mutation, such mutations have a much smaller mutational target size and are therefore much less frequent. Moreover, one might wonder why it is specifically Bem3, rather than one of the other GAPs, that is lost to rescue the *bem1Δ* strain. Some hints to answer this outstanding question are provided by a detailed theoretical analysis of the rescue mechanism later in the section "Functional submodules of cell polarization".

### Copy number variation experiments confirm theoretical predictions

Based on the GAP/Cdc42-ratio constraint in the rescue mechanism, our theory makes two specific predictions: (i) Increasing the protein copy number (i.e. overexpression) of Cdc42 will rescue cell polarization of *bem1Δ* cells by invoking the Bem1-independent mechanism. (ii) Polarization of *bem1Δbem3Δ* cells will break down if the protein copy number of Cdc42 is lowered compared to the WT level (Fig. 2b).

To test these model predictions experimentally, we first constructed different yeast strains with Cdc42, labeled with sfGFP, under an inducible galactose promoter. This allows us to tune the Cdc42 protein copy number by varying the galactose concentration in the growth media[37]: a *bem1Δ* strain (yWKD069), a *bem1Δ bem3Δ* (yWKD070), and a modified WT strain (yWKD065) (see Methods). We confirmed that the sfGFP tag on our inducible Cdc42 does not significantly alter fitness (see Supplementary Note 6.2), in line with literature on viability and localization of another fluorescent Cdc42 sandwich fusion[38] in budding yeast. As a next step, we inoculated the different strains at varying galactose concentration in 96 well plates, that were placed in a plate reader to measure the cell density over time, and thereby determined the population growth rate (see Methods). For every galactose concentration, the growth rates are normalized to those of WT cells, with Cdc42 under its native promotor (yLL3a), grown at the same galactose concentration. In Fig. 3a the normalized growth rates of the different mutants are plotted. As expected, WT cells grow at all galactose concentrations. In contrast, WT cells with Cdc42 under the galactose promotor (yWKD065), do not grow in the absence of Cdc42 (0% galactose concentration), since a failure to polarize severely impairs cell division and eventually leads to cell death and thus zero growth rate[8]. Our data show that the WT mechanism is rather insensitive to Cdc42 protein copy number, even for very low expression of Cdc42, in accordance with theory (Fig. 2a).

Our model predicts that *bem1Δ* cells need the highest Cdc42 protein copy number to polarize, WT cells will need the least, and the *bem1Δ bem3Δ* cells should be in between. We indeed find that the *bem1Δ* strain (yWKD069) grows in media with 0.1% or higher galactose concentration. We inoculated these strains at lower galactose concentrations, but never observed growth for the *bem1Δ and* the *bem1Δbem3Δ* strains in more than one technical replicate (out of 6 and 4 respectively) per condition (see Supplementary Table S3). We attribute the rare growth at low galactose concentrations to emergence of suppressor mutations. Therefore, we focus on comparing growth rates. There is strong and positive evidence that the *bem1Δbem3Δ* grows faster than the *bem1Δ* in the 0.06,% 0.1% and 0.2% galactose concentration respectively (Bayes factors 7, 131 and 6, and using interpretation qualifications from ref. 39). For WT cells with Cdc42 under the galactose promotor we observe and reduced growth rate at 0.01% galactose concentration but growth is only fully inhibited at 0% galactose concentration. All of the above experimental observations agree with our specific theoretical predictions.

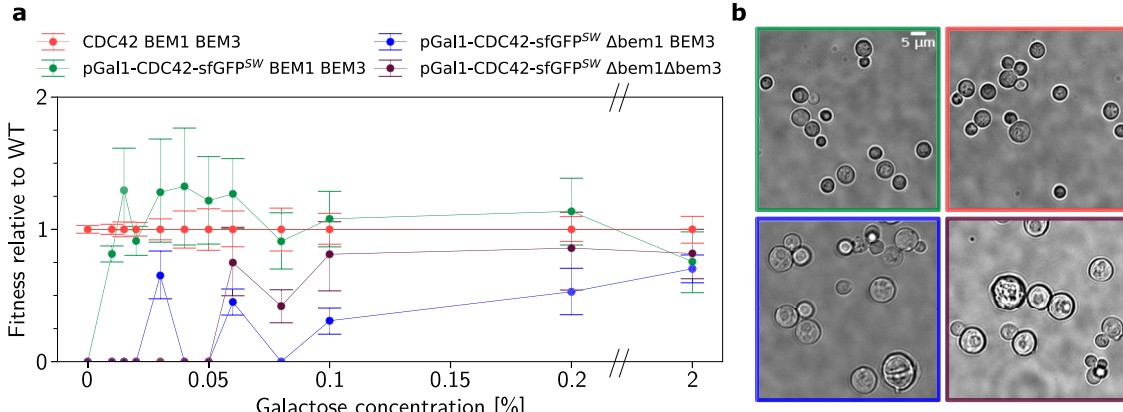

**Fig. 3 | Experiments confirm theoretically predicted effect of Cdc42 protein copy number on the latent polarity-mechanism. a** Relative growth rate (fitness) of mutants as a function of the galactose concentration (proxy for Cdc42 protein copy number) show that higher expression of Cdc42 rescues *bem1Δ* cells and to a lesser extent *bem1Δ bem3Δ* cells. Markers denote the means of the posterior probability distribution for the fitness, the error bars indicate the 68% credible intervals (see Methods). Large error bars result from variability between technical replicates and because in some conditions, e.g. *bem1Δ* at 0.03% galactose, growth is very infrequent (see Supplementary Note 6.1). The number of experiments per strain-condition pair is given in Supplementary Table S5. For the absolute growth rates (not normalized to WT), see Supplementary Fig. S6. **b** Micrographs of all strains in 0.06% galactose, after 24 h of incubation. WT yeast cells with Cdc42 under the galactose and native promotor respectively are entering stationary phase and diluted 1000× (top row). The *bem1Δ* and *bem1Δ bem3Δ* cells are in log phase and diluted 100× (bottom row).

Furthermore, we examined the influence of Cdc42 protein copy number on cell morphology (see Fig. 3b) and viability as discussed in the Supplementary Material. These experiments provide support for the conclusions from our growth assays, namely that viability increases and size (as proxy of polarization time[3,40]) decreases with increasing protein copy number.

Taken together, the experimental data confirm the theoretical prediction that the Bem1-independent rescue mechanism is operational only below a threshold GAP/Cdc42-concentration ratio. In addition, we find that the Bem1-dependent WT mechanism is surprisingly insensitive to Cdc42 protein copy number, i.e., operates also at very low Cdc42 concentration. In the context of our theory, this significant difference in Cdc42 protein copy number sensitivity is explained by the qualitative difference of their principles of operation (see "The Cdc42 interaction network facilitates a latent polarization mechanism"). The WT mechanism is based on recruitment of the GEF Cdc24 to the polar zone, mediated by the scaffold protein Bem1. In contrast, the rescue mechanism crucially involves enzyme saturation of Cdc42 hydrolysis due to high Cdc42 density in the polar zone. This enzyme saturation requires a sufficiently large Cdc42 protein copy number relative to the GAP protein copy number. In the section "Functional submodules of cell polarization" below, we will analyze the mathematical model, and the qualitative and conceptual differences between these two mechanisms in more detail.

### The latent rescue mechanism explains and reconciles previous experimental findings

In previous experiments, several Bem1 mutants were studied that perturb Bem1's ability to mediate co-localization of Cdc24 to Cdc42-GTP, the key feature that underlies operation of the WT mechanism[17,20,21,41–43]. The observations from these experiments have remained puzzling and apparently conflicting among one another as of yet. As we show in detail in the Supplementary Discussion in Supplementary Note 5, the latent rescue mechanism predicted by our mathematical model explains and reconciles all of these previous experimental findings. The key insight is that the latent rescue mechanism can be activated by a global increase of GEF activity (see dashed line in Fig. 2b). Bem1 mutants that lack the Cdc42-interaction domain but still bind to the GEF Cdc24 may provide such a global increase of GEF activity and thus rescue polarization of *bem1Δ* cells.

Moreover, in accordance with optogenetics experiments[43], our mathematical model predicts that the latent Bem1-independent mechanism can also be induced outside the regime of spontaneous polarization by a sufficiently strong local perturbation of the membrane-bound GEF concentration.

### Functional submodules of cell polarization

Cell polarization in budding yeast is a functional module based on a complex protein interaction network with Cdc42 as the central polarity protein (cf. Fig. 1b–d). As we discuss next, the full network can be dissected into *functional submodules*. Here, the term functional submodule refers to a *part* of the full interaction network with a well-defined function in one or more pattern-forming mechanisms. Our theoretical analysis will reveal that an interplay of two (or more) *functional submodules* each constitutes a fully functional cell polarization mechanism. Importantly, the submodules *emerge* from the interplay of various players (components) in the biochemical interaction network and the spatial transport of proteins (by diffusion and along actin cables).

As we argued in the "Introduction", establishment and maintenance of cell polarity requires that Cdc42-activity is localized to membrane regions with a high density of Cdc42. This can be achieved in two different ways. First, by the recruitment of the scaffold protein Bem1 to Cdc42-GTP, which in turn recruits the GEF (Cdc24) and thus localizes Cdc42 activation to the polar zone, where Cdc42 density is high (Fig. 4a, top left). We call this the *polar activation* submodule. Second, GAP saturation in regions of high local Cdc42 densities can localize Cdc42 activity to the polar zone (Fig. 4a, top right), as described above in the subsection "GAP saturation can localize Cdc42 to the polar zone". The transient (partial) sequestration of GAPs in Cdc42-GAP complexes is essential for this *polar GAP saturation* submodule. The third submodule (Fig. 4a, bottom) that we term *Cdc42 transport*, comprises various modes of Cdc42 transport towards the polar zone: vesicle transport along polarized actin cables (cf. Fig. 1b) and effective (self-)recruitment of Cdc42 from the cytosol. Several experiments indicate that downstream effectors of active Cdc42, such as Cla4, Gic1 and Gic2 may provide such effective recruitment in the absence of Bem1[31,33,44].

These three functional submodules represent different mechanistic aspects of the Cdc42-interaction network. Each submodule is

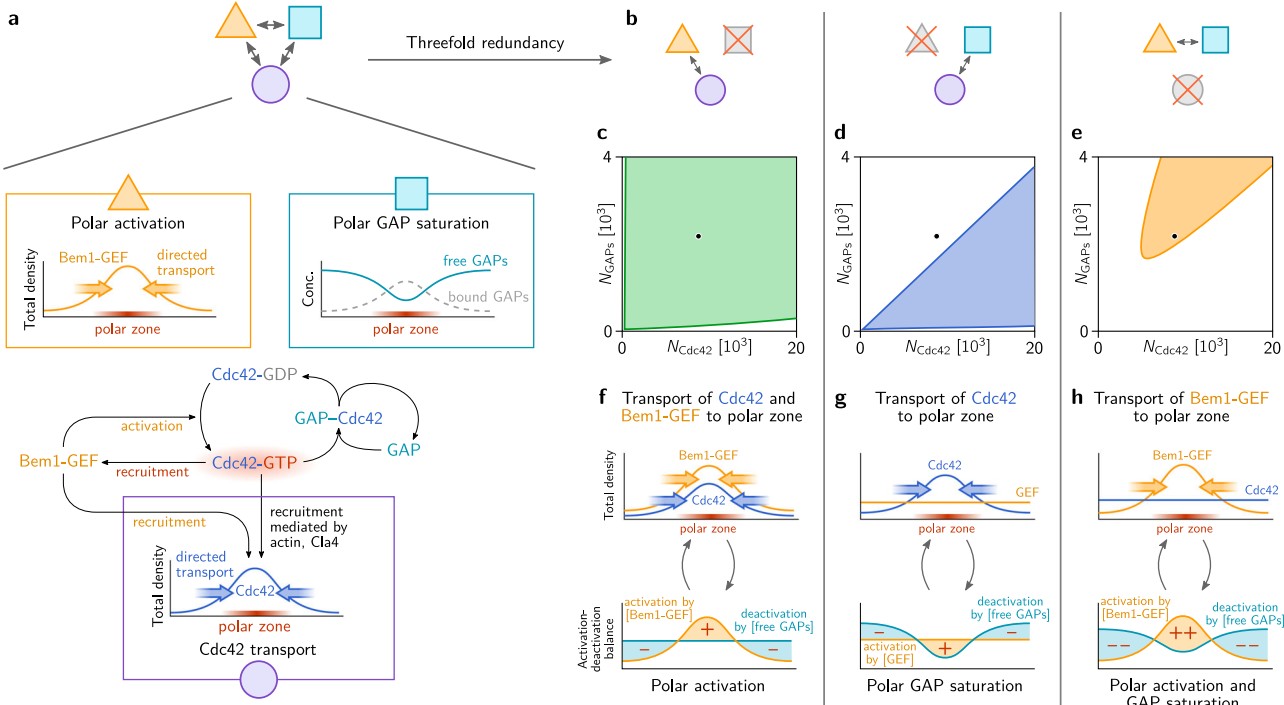

**Fig. 4 | Three functional submodules constitute three distinct mechanisms of Cdc42-GTP polarization. a** Three functional submodules of the Cdc42 interaction network contribute to the formation and maintenance of a *polar zone* (region of high Cdc42-GTP concentration, highlighted in red): Transport of Cdc42 towards the polar zone (purple circle). High Cdc42 activity can be maintained due to GAP saturation in the polar zone (teal square) and by transport of the GEF to the polar zone via the scaffold protein Bem1 (yellow triangle). **b** Combinations of pairs of these functional submodules constitute mechanisms of self-organized pattern formation. **c–e** These mechanisms are operational in different regimes of the total protein copy number of Cdc42 and GAPs. The WT mechanism (**f**) is largely insensitive to protein copy number variations (**c**) because it is based on mutual recruitment of Cdc42 and Bem1-GEF complexes, and does not depend on saturation of GAPs in the polar zone. In contrast, when the GEF is not transported to the polar zone (e.g. due to a deletion of Bem1), only GAP saturation in the polar zone maintains high Cdc42 activity there, while deactivation dominates away from the polar zone. Therefore, the polarization mechanism (**g**) is sensitive to the GAP protein copy number (**d**). **h** Remarkably, if transport of Cdc42 is suppressed, e.g. by strongly binding it to the membrane, a combination of Bem1-GEF complex recruitment and polar GAP saturation maintain a localized high Cdc42 activity even though the total density of Cdc42 is homogenously distributed.

operational only under specific constraints on the biochemical properties and protein copy numbers of the involved proteins. In the following, we exploit these constraints to study the roles of the submodules in the mathematical model by disabling them one at a time. This allows us to tease apart the mechanisms that are operational under the corresponding experimental conditions. The first submodule, *polar activation*, is disabled by the knock-out of Bem1. The second submodule, *polar GAP saturation*, is suppressed if the protein copy number of GAPs is too high. Alternatively, polar GAP saturation is rendered non-operational if the dissociation rate of the GAP-Cdc42 complex is too fast, or if the free GAPs diffuse very fast making additional free GAPs readily available in the polar zone. The third submodule, *Cdc42 transport*, can be switched off by immobilizing Cdc42, i.e., suppressing its spatial redistribution. Experimentally, this has been achieved in fission yeast by fusing Cdc42 to a transmembrane protein that strongly binds to the membrane and is nearly immobile there[41].

It is worth noting that Bem1 is part of two functional submodules: Recruiting GEF to the polar zone provides *polar activation*, recruiting Cdc42 contributes to *Cdc42 transport*. While polar activation is entirely dependent on Bem1, there are several Bem1-independent modes of Cdc42 transport, including actin-based vesicle trafficking and other putative recruitment mechanisms (cf. Fig. 1d). Thus, the Cdc42 transport submodule is still operational in *bem1Δ* cells.

We next performed linear stability analysis for the full mathematical model under each of these perturbations disabling one of the submodules at a time (as described in detail in Supplementary Note 4; see Supplementary Table S3). In each case we found that the remaining two submodules operate in concert to constitute a mechanism for

spontaneous Cdc42 polarization as illustrated in Fig. 4b. Figure 4c–e shows the regime of operation of the three different mechanisms as a function of the total Cdc42 and GAP concentrations. Figure 4f–h illustrate the concerted interplay of directed protein-transport and regulation of Cdc42 activity (activation/deactivation) that underlie Cdc42-polarization in these three mechanisms.

Before we turn to the detailed descriptions of these mechanisms, we note that if two submodules are disabled simultaneously, the remaining submodule alone cannot facilitate pattern formation. In particular, and perhaps somewhat counterintuitively, self-recruitment of Cdc42 alone is not sufficient to drive spontaneous cell polarization[34,45].

**Wild-type mechanism: Cdc42 transport plus polar activation.** The interplay of the Cdc42 transport submodule and the Cdc42-Bem1-Cdc24 recruitment submodule (polar activation), illustrated in Fig. 4f, constitutes the WT mechanism that operates via mutual recruitment of Cdc42 and Bem1[8,11,12]. Characteristic for this mechanism is the co-localization of Cdc24 and Cdc42-GTP in the polar zone, as observed in previous experiments[42,43]. Other than the rescue mechanism, the mutual recruitment mechanism does not require polar GAP saturation. Therefore, it is insensitive against high concentration of GAPs, i.e., it is operational for much higher GAP/Cdc42-concentration ratios than the rescue mechanism. Furthermore, it is robust against high diffusivity of free GAPs and high catalytic rates of the GAPs (fast decay of GAP-Cdc42 complexes into free GAP and Cdc42-GDP). This implies that in mathematical models of the WT mechanism the GAPs can be accounted for *implicitly* by a constant and homogeneous hydrolysis rate, as in

previous models[10,12,42,46]. Notably Bem1 mediates both polar activation and Cdc42 transport (via recruitment from the cytosol) in these models.

**Rescue mechanism: Cdc42 transport plus polar GAP saturation.** The latent, Bem1-independent rescue mechanism operates by the interplay of GAP saturation in the polar zone (illustrated in Fig. 4g) and Cdc42 transport (including effective self-recruitment via actin and/or other downstream effectors like Cla4). Characteristic for this mechanism is that it does not require co-localization of Cdc24 to Cdc42-GTP in the polar zone (see Fig. 4g). In future experiments, this lack (or strong reduction) of Cdc24 polarization could serve as a clear indicator of the rescue mechanism. As explained above, the rescue mechanism relies on GAP saturation in the polar zone to maintain high Cdc42 activity there. When Cdc42 activity is maintained by lower GAP activity, we expect longer residence times of Cdc42 in the polar zone compared to WT cells. This prediction could be tested in future experiments.

GAP saturation is suppressed by either high abundance, high catalytic activity, or fast transport (by cytosolic diffusion or vesicle recycling) of the GAPs. The last constraint provides a plausible explanation why it is specifically Bem3 that needs to be deleted to rescue *bem1Δ* cells. In contrast to Rga1 and Rga2, Bem3 has been found to be highly mobile, probably because it cycles through the cytosol[47]. GAP saturation, i.e. the depletion of free GAPs in the polar zone, entails a gradient of the free GAP density towards the polar zone. A mobile GAP species like Bem3 will quickly diffuse along this gradient to replenish the free GAPs in the polar zone. This influx relieves the GAP saturation there and thus counteracts the activation of Cdc42 in the incipient polar zone. Therefore, the loss of Bem3, rather than one of the other, less mobile GAPs, promotes the formation of a stable polar zone.

**Polarization with immobile Cdc42: Bem1-mediated recruitment plus polar GAP saturation.** The interplay of Cdc42-Bem1-Cdc24 recruitment (polar activation) and the polar GAP saturation, illustrated in Fig. 3H, facilitates polarization of Cdc42 activity without the spatial redistribution Cdc42's total density (blue line in Fig. 3h, top). Instead, the proteins that are being redistributed are Bem1 and GEF. The polar zone is characterized by a high concentration of membrane-bound Bem1–GEF complexes which locally increase Cdc42 activity. Cdc42-GTP, in turn, recruits further Bem1 and GEF molecules to the polar zone. Characteristic for this mechanism is that Cdc42-GTP is polarized while the total Cdc42 density remains uniform on the membrane. Experimentally, this has been observed in fission yeast using Cdc42 fused to a transmembrane domain (Cdc42-psy1^TM) that renders Cdc42 nearly immobile. The polarization machinery of fission yeast is closely related to the one of budding yeast; it operates based on the same mutual recruitment pathway with Scd1 and Scd2 taking the roles of Cdc24 and Bem1[14]. In future experiments, it would be interesting to test whether the Cdc42-psy1^TM also facilitates polarization in budding yeast (potentially in a strain with modified GAP or Cdc42 protein copy number as the regime of operation might not coincide with the WT protein copy numbers).

## Discussion

### Mechanistic understanding of the cell polarization module in budding yeast

We have discovered that multiple, redundant self-organization mechanisms coexist within the protein network underlying cell polarization in budding yeast. This explains the remarkable resilience of this module: It remains operational under many experimental (genetic) perturbations[13,20,21,41,43,48]. While we find that the Cdc42-polarization machinery is robust against many genetic perturbations, we have put particular focus on one of its key components, Bem1, since a previous experiment has found quick and reproducible recovery from its deletion[3]. By dissecting the full cellular polarization module

into *functional submodules*, we have identified three distinct mechanisms of self-organized pattern formation. Besides the wild-type mechanism relying on the colocalization of Cdc42 with its GEF via Bem1, this includes a latent and Bem1-independent rescue mechanism and a mechanism that is independent of Cdc42 redistribution. Our theory, which is compatible with published experiments, reveals that these mechanisms share many components and interaction pathways of this network. This implies that the redundancy of cell polarization is not at the level of individual components or interactions but arises on the level of the emergent function itself. If one submodule is rendered non-functional, the combination of the remaining submodules still constitutes an operational mechanism of cell polarization − if parameters, in particular protein copy numbers, are tuned to a parameter regime where these remaining submodules are operational. Redundancy hence provides adaptability − the ability to maintain function despite (genetic) perturbations. Importantly, the submodules are emergent: they involve the interplay of several network components, their biochemical interactions, and their spatial transport.

Our analysis in terms of functional submodules provides a mechanistic understanding of the polarization machinery where molecular details have been "coarse grained". In the context of genotype–phenotype maps, this coarse-grained description could be integrated with into a cell cycle model to address questions about epistasis[49], and eventually predict evolutionary trajectories in a population dynamics model.

Interestingly, the formation of Min-protein patterns in *E. coli* relies on the same type of mechanism as the rescue mechanism for Cdc42-polarization: self-recruitment of an ATPase (MinD) and enzyme saturation of the AAP (MinE) that catalyzes MinD's hydrolysis and subsequent membrane dissociation[50–52]. The transient MinDE complexes play the analogous role to the Cdc42-GAP complexes here: In regions of high MinD density, MinE is sequestered in MinDE complexes, which limits the rate of hydrolysis until the complexes dissociate or additional MinE comes in by diffusion. Because MinE cycles through the cytosol, it rapidly diffuses into the polar zone where the density of free MinE is low. This diffusive influx relieves the enzyme saturation in the polar zone and eventually leads to a reversal of the MinD polarity direction. The repeated switching of MinD polarity due to redistribution of MinE is what gives rise to the Min oscillations in *E. coli*. Recently also stationary Min patterns have been observed in vitro[53]. Conversely, oscillatory Cdc42 dynamics are found in the fission yeast *S. Pombe*[32], and have also been indirectly observed in budding yeast mutants[46,54].

### The physics of self-organization imposes constraints on evolution

The fundamental question of evolutionary cell biology is "How do cells work and how did they come to be the way they are?"[55]. Our in-depth analysis of the yeast polarization machinery gives an answer to the first half of this question for a specific biological system. It also allows us to approach the second half and develop a concrete hypothesis how the Cdc42 cell-polarization machinery of budding yeast might have evolved from a more rudimental ancestral form.

Our theoretical and experimental results highlight the importance of protein copy numbers as control parameters that determine whether a mechanism of spontaneous cell polarization is operational. Phrased from a genetic perspective, the genes that code for components of the cell polarization machinery are *dosage sensitive*[56]. On the one hand, this entails that mutations of cis-regulatory elements (like promoters and enhancers)[57] can tune the protein copy numbers of proteins to the regime of operation of a specific cell-polarization mechanism and optimize the function within that regime. On the other hand, protein copy number sensitivity constrains evolution of the polarization-machinery's components via duplication and sub-functionalization[56,58].

One of our key findings is that the constraints on a single particular mechanism can be circumvented by the coexistence of several redundant mechanisms of self-organization that operate within the same protein-interaction network. The regimes of operation − and, hence the dosage sensitivity of specific genes − can differ vastly between these distinct mechanisms. Therefore, redundancy on the level of mechanisms allows the module's components to overcome constraints like protein copy number sensitivity and thus promotes "evolvability" − the potential of components to acquire new (sub-) functions while maintaining the module's original function. Previous work has shown how additional negative feedback loops can also increase the regime of operation of WT mechanism[28].

A particular example in budding yeast's cell-polarization module where duplication and sub-functionalization might have taken place is the diversification of the different GAPs of Cdc42 in budding yeast. Bem3, Rga1, and Rga2 play individual roles in specific cellular functions, like the pheromone response pathway[47,59], axial budding[60], and the timing of polarization[61]; see[62] for a visualization. At the origin of this diversity of GAPs is its promotion by cell-polarization mechanisms that are insensitive to GAP protein copy number, such as the Bem1-mediated WT mechanism. As we will argue below, this notion provides a concrete hypothesis about the role of scaffold proteins, like Bem1, for the evolution of functional modules that operate by the interplay of many interacting components.

## How evolution might leverage scaffold proteins

In the context of cellular signaling processes it was suggested previously that evolution might leverage scaffold proteins to evolve new functions for ancestral proteins by regulating selectivity in pathways, shaping output behaviors and achieving new responses from pre-existing signaling components[63]. Our study of the Cdc42 polarization machinery gives a perspective on how scaffold proteins may also play an important role in the evolution of intracellular self-organization. The scaffold protein Bem1 − by connecting Cdc42-GTP to Cdc42's GEF − generates a functional submodule that contributes to self-organized Cdc42 polarization. Based on this, we propose a hypothetical evolutionary history for Bem1, illustrated in Fig. 5: The latent rescue mechanism is generic and rudimentary and therefore might be an ancestral mechanism of Cdc42 polarization in fungi. On this basis, Bem1 could then have evolved in a step-wise fashion: A hypothetical Bem1 precursor binding to Cdc24 but not to Cdc42-GTP might have facilitated a globally enhanced catalytic activity of Cdc24 by relieving its auto-inhibition[22,23]. Our theory shows that such an increase of GEF activity enlarges the range of GAP/Cdc42-concentration ratios for which the latent rescue mechanism is operational. This would have entailed an evolutionary advantage by increasing the robustness of the (hypothetical) ancestral mechanism against protein copy number variations. In a subsequent step the Bem1-precursor might then have gained the Cdc42-binding domain (SH3 domain) by domain fusion[64], thus forming the full scaffold protein that connects Cdc24 to Cdc42-GTP that mediates the WT polarization mechanism (mutual recruitment of Cdc24 and Cdc42). Along this hypothetical evolutionary trajectory, the constraints on the GAP/Cdc42 protein copy number ratio and the molecular properties of the GAPs (kinetic rates, membrane affinities) would be relaxed, thereby allowing the duplication and sub-functionalization of the GAPs[58]. Given that Bem1 is highly conserved in fungi[2], and that fission yeast polarization is based on the same mutual recruitment mechanism[65,66], this hypothetical evolutionary pathway might lie far in the past.

There are several possible routes to test our hypotheses. One possibility is the construction of phylogenetic trees for the different proteins (domains) that could inform on the order they appeared

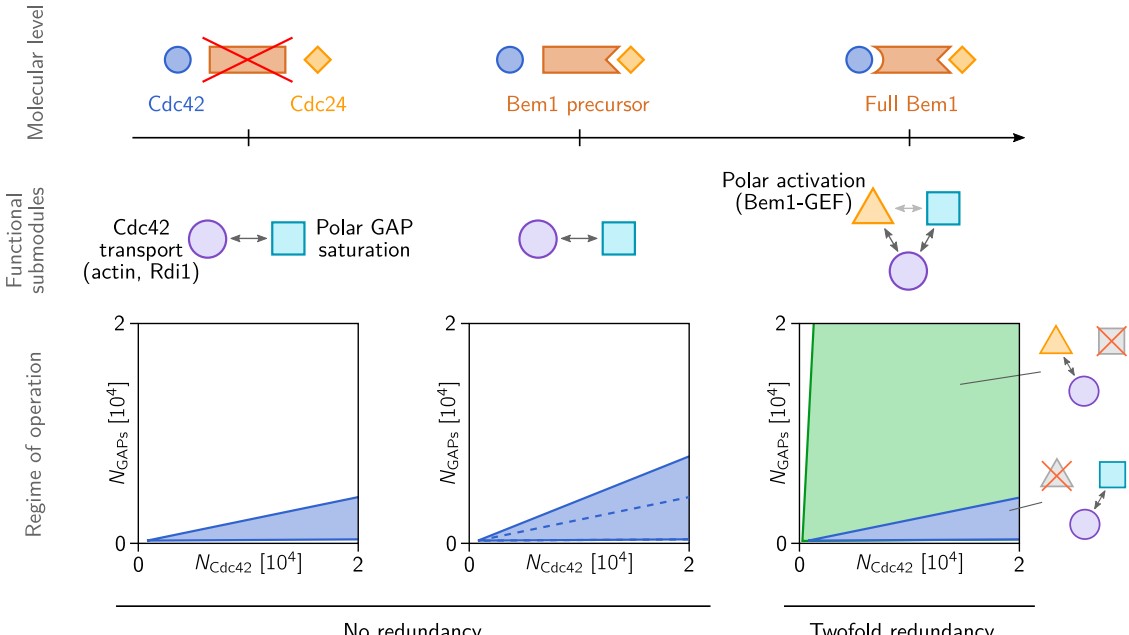

**Fig. 5 | Hypothetical evolution of Bem1.** (*Left*) The Bem1-independent "rescue" mechanism based on GAP saturation and Cdc42 transport towards membrane bound Cdc42-GTP is operational only in a limited range of the GAP/Cdc42-concentration ratios (cf. Fig. 4d). (*Center*) a Bem1 precursor (Bem1-fragment) that binds to Cdc24 and relieves its auto-inhibition increases the range of viable GAP/Cdc42-concentration ratios and thus increases the robustness against protein copy number variations (cf. Fig. 2). It does, however, not change the underlying mechanism qualitatively. (*Right*) Domain fusion of a Cdc42-GTP-binding domain with the Cdc24-binding Bem1-precursor, leads to a new connection in the Cdc42-interaction network that leads to recruitment of Cdc24 to the polar zone. On the level of submodules, this new connection constitutes a new functional submodule that we called "polar activation" (yellow triangle). In conjunction with transport of Cdc42 towards the polar zone, polar activation gives rise to the highly robust mutual-recruitment mechanism that is operational in WT yeast (regime of operation shaded in green in the ($N_D$, $N_G$)-parameter plane; cf. Fig. 4c). Note that the scale on the vertical axis is chosen larger to emphasize the significantly larger regime of operation of the Bem1-mediated mechanism.

during evolution of the polarity network[67]. Another possibility is to search for species in the current tree of life which contain intermediate steps of the evolutionary trajectory. For instance, species with a more ancient version of Bem1 lacking the SH3 domain, and identify the protein self-organization principles underlying polarization in these species. This is becoming a more and more realistic option, given the very large (and still expanding) number of fungal species that has been sequenced[2] and the growing interest of cell and molecular biologists to work with non-model systems[68].

On a broader perspective, we have shown how understanding the mechanistic principles underlying self-organization can provide insight into the evolution of cellular functions, a central theme in evolutionary cell biology. Specifically, we have presented a concrete example that shows how a self-organizing system might have evolve from more a rudimentary, generic mechanism that is parameter sensitive, to a specific, robust and tightly controlled mechanism by only incremental changes[69].

## Methods
### Model motivation and assumptions
The primary goal of the mathematical model we propose is to explain the rescue of *bem1Δ* cells by the loss of *BEM3*. To that end, we make minimal, but essential extensions to a previously established model[12] that accounts for the core Cdc42-polarization mechanism relying on the Bem1-mediated pathway[8,14,18,41,43]. Importantly, the extended model we propose here enables us to explain several previous experimental findings that had remained puzzling so far. We will summarize and discuss these findings that serve as additional support for our model in the supplementary discussion (Supplementary Note 5). In what follows, we describe the biophysical and biochemical processes (diffusion, vesicle-based transport and protein interactions) accounted for by our model. The mathematical formulation of the model in the framework of bulk-surface coupled reaction–diffusion systems is presented in the subsequent section. Linear stability analysis, parameter sampling and numerical simulations are described in Supplementary Notes 2–4.

**Protein transport membrane-recruitment.** The cell is modeled as a spherical domain with a diffusive bulk (cytosol) on the inside and the membrane on the surface where proteins interact and diffuse laterally (see Supplementary Fig. 1). Bulk and surface dynamics are coupled due to membrane attachment and detachment of proteins. Mathematically, the model is formulated as a reaction--diffusion system with bulk-surface coupling. As we will argue in the following, both transport pathways – cytosolic cycling and vesicle-based transport (Fig. 1b, c in the main text) – can be incorporated in this modeling framework.

In previous works, vesicle-trafficking along actin cables has been modeled to various degrees of detail and based on different assumptions[9,11,29,30,70–72]. However, a mechanistically detailed modeling of vesicle trafficking is not feasible at the moment because the highly complex vesicle recycling pathway – involving endocytosis, transport along actin cables, processing in intracellular membrane compartments like endosomes and the Golgi apparatus, and finally exocytosis – is not fully characterized experimentally. As we will see, however, a detailed description is not required for the purpose of the analysis here. Instead, we model vesicle recycling of Cdc42 as effective membrane-recruitment of Cdc42-GDP by Cdc42-GTP. This effective description incorporates the two essential features of vesicle recycling that are relevant for the polarization machinery: (i) vesicle transport is directed towards membrane-bound Cdc42-GTP and (ii) Cdc42 delivered to the membrane by vesicles upon exocytosis is (mostly) GDP-bound[71]. Details are discussed in Supplementary Note 1.

In addition to vesicle recycling, several downstream effectors of Cdc42-GTP – Cla4, Gic1/Gic2, and flippase[31,32,48,73,74] – have been suggested to facilitate membrane-recruitment of Cdc42-GDP (see Supplementary Note 1 for details). We incorporate these putative Cdc42-GDP-recruitment pathways together with vesicle-based Cdc42-transport by a single, effective recruitment process that is directed by membrane-bound Cdc42-GTP (illustrated in Fig. 1d (4) in the main text).

**Biochemical interactions.** Figure 1D shows the biochemical interaction network underlying our model. At its core is the GTPase cycle of Cdc42 ((1) in Fig. 1d). Cdc42 cycles between an active, GTP-bound, and an inactive, GDP-bound state on the membrane. In its GDP-bound form, Cdc42 can bind to the guanine-nucleotide-dissociation inhibitor (GDI) Rdi1, which sequesters Cdc42's membrane binding anchor and thus enables it to diffusive freely in the cytosol. The cycling of Cdc42 between its GTP- and GDP-bound states is regulated by the guanine nucleotide-exchange factor (GEF) Cdc24, and GTPase-activating proteins (GAPs) that catalyze the hydrolysis from GDP to GTP. In wild-type cells, Cdc42-GTP recruits the scaffold protein Bem1 to the membrane which in turn recruits the GEF Cdc24 ((2) in Fig. 1d) to form a Bem1–GEF complex. These membrane-bound Bem1–GEF complexes recruit Cdc42-GDP from the cytosol to the membrane and activate it there ((3) in Fig. 1d), thus closing the feedback loop (mutual recruitment) that underlies WT polarity[8,10,15]; see refs. 14 and [52] for recent reviews through the experimental and theoretical lens, respectively. This feedback loop is captured by the model introduced in ref. 12. We make the following key extensions:

- Explicit modeling of Cdc42's hydrolysis by GAPs as a catalytic reaction with an intermediate Cdc42-GAP complex[25].
- Effective membrane-recruitment by membrane-bound Cdc42-GTP, accounting for vesicle-based transport of Cdc42 towards zones of high Cdc42-GTP concentration as well as further putative recruitment pathways mediated by downstream effectors of Cdc42-GTP[21,31–33].
- Membrane binding of the GEF Cdc24 independently of Bem1 via Cdc24's PH domain[75].

The model analysis in terms of functional subunits shows that all three extensions are required to describe the rescue of *bem1Δ* mutants in the model. Further details of these model extensions, biological motivation and the underlying assumptions are discussed in the Supplementary Note 1.

### Reaction–diffusion dynamics with bulk-surface coupling
**General framework.** Since budding yeast cells are (nearly) spherical, we study the proteins' reaction–diffusion dynamics in a spherical geometry composed of a cytosol (bulk) of radius $R$ with membrane on its surface (Supplementary Fig. S1). Naturally, we choose spherical coordinates $(r,\varphi,\theta)$. For a general, compact notation, we denote concentrations of membrane-bound and cytosolic components by vectors $m$ and $c$, respectively.

In the bulk, we consider purely diffusive dynamics,

$$\partial_t c(r,\varphi,\theta,t) = D_c \nabla^2 c, \tag{1}$$

with the matrix of diffusion constants $D_c = \mathrm{diag}(\{D_i\})$. Unless stated otherwise, the cytosolic diffusion constants are all set to the same value $D_c$ such that $D_c = D_c$.

In spherical coordinates, the Laplacian $\nabla^2$ acting on some function $\psi$ reads

$$\nabla^2 \psi = r^{-2}\partial_r(r^2\partial_r \psi) + \nabla_S^2 \psi, \tag{2}$$

**Table 1 | Variables of the reaction--diffusion model describing the protein concentrations of Cdc42, GAPs, Bem1 and GEF (Cdc24) in various conformational states – cytosolic, membrane bound and in form of multi-protein complexes**

| Domain [Unit] | Symbol | Description |
|---|---|---|
| Cytosol [$\mu m^{-3}$] | $c_D$ | Cdc42-GDP (potentially GDI-bound) |
| | $c_B$ | Bem1 |
| | $c_F$ | GEF |
| Membrane [$\mu m^{-2}$] | $m_d$ | Free Cdc42-GDP |
| | $m_t$ | Cdc42-GTP |
| | $m_g$ | GAP |
| | $m_{tg}$ | Heterodimeric Cdc42-GAP complexes |
| | $m_{bf}$ | Bem1 |
| | $m_{bf}$ | Heterodimeric Bem1-GEF complexes |
| | $m_f$ | GEF |

For descriptions of the model parameters see Supplementary Table S1.

where the "angular" Laplacian on the sphere's surface $S$ is given by

$$\nabla_S^2 \psi = \frac{1}{r^2 \sin(\theta)} \left[ \partial_\theta (\sin(\theta) \partial_\theta \psi) + \partial_\varphi^2 \psi \right]. \quad (3)$$

The bulk is coupled to the membrane by attachment-detachment reactions that lead to bulk flows, $f$, normal to the surface

$$-D_c n \cdot \nabla c|_{r=R} = f\left( m, c|_{(r=R)} \right), \quad (4)$$

where $n$ is the surface's inward normal vector. In spherical coordinates, the radial gradient is given by the radial derivative $n \cdot \nabla = -\partial_r$. The attachment-detachment flows $f$ of our specific model will be specified further below.

The dynamics of membrane-bound components are given by

$$\partial_t m(\varphi, \theta, t) = D_m \nabla_m^2 m + g\left( m, c|_{r=R} \right), \quad (5)$$

where the nonlinear function $g$ encodes the nonlinear reactions on the membrane. Note that the diffusion operator on the membrane $\nabla_m^2 = \nabla_S^2|_{r=R}$ coincides with the bulk Laplacian $\nabla^2$ restricted to the membrane at $r=R$. This is because the sphere fulfills the rotational symmetries of the diffusion operator.

**Variables, reaction terms, and conserved protein numbers.** As shorthands for the protein concentrations we use the same shorthand notation as in Fig. 1 in the paper: D – Cdc42-GDP; T – Cdc42-GTP; G – GAPs; B – Bem1; F – GEF. (Note that we refer to Cdc24 as GEF to prevent confusion with Cdc42.) We denote concentrations of membrane-bound species with the symbol $m$ with lowercase subindices, and cytosolic concentrations using the symbol $c$ with uppercase subindices (see Table 1). Using the vector notation introduced above, we have $c = (c_D, c_B, c_F)$, $m = \left( m_d, m_t, m_{tg}, m_g, m_b, m_{bf}, m_f \right)$.

The protein interactions described in the results section and illustrated in Fig. 1 are modeled by mass-action law kinetics, with the reaction rates described in Supplementary Table S1. The reaction kinetics read

$$f(m,c) = \begin{pmatrix} k_d m_d - \left( k_D + k_{tD} m_t + k_{bfD} m_{bfD} m_{bf} \right) c_D \\ k_b m_b - k_{tB} m_t c_B \\ k_f m_f + k_{bf} m_{bf} - \left( k_F + k_{bF} m_b \right) c_F \end{pmatrix}, \quad (6)$$

And

$$g(m,c) = \begin{pmatrix} (k_D + k_{tD} m_t) c_D + k_{tg} m_{tg} - \left( k_{fd} m_f + k_{bfd} m_{bf} + k_d \right) m_d \\ \left( k_{fd} m_f + k_{bfd} m_{bf} \right) m_d + k_{bfD} m_{bf} c_D - k_{tg} m_t m_g \\ k_{tg} m_t m_g - k_{gt} m_{tg} \\ -k_{tg} m_t m_g + k_{gt} m_{tg} \\ k_b m_t c_B - k_b m_b + k_{bf} m_{bf} - k_{bF} m_{bf} c_F \\ k_{bF} m_{bf} c_F - k_{bf} m_{bf} \\ k_F c_F - k_f m_f \end{pmatrix}. \quad (7)$$

A more detailed discussion of the assumptions and motivation for the specific terms is provided in Supplementary Note 1. The above reaction−diffusion dynamics conserve the total numbers of Cdc42, GAPs, Bem1, and GEF molecules,

$$N_{Cdc42} = \int_V d^3 x\, c_D + \int_S d^2 \sigma (m_d + m_t + m_{tg}), \quad (8)$$

$$N_{GAPs} = \int_S d^2 \sigma (m_g + m_{tg}), \quad (9)$$

$$N_{Bem1} = \int_V d^3 x\, c_B + \int_S d^2 \sigma (m_b + m_{bf}), \quad (10)$$

$$N_{GEF} = \int_V d^3 x\, c_F + \int_S d^2 \sigma (m_f + m_{bf}). \quad (11)$$

Hence, these protein copy numbers are *control parameters* of the model.

**Model analysis.** We analyzed the above reaction−diffusion equations using linear stability analysis and numerical simulations. Details are provided in the Supplementary Notes 2–4. In brief, linear stability analysis yields the growth rates of perturbations of the homogeneous steady state. From the resulting dispersion relation (illustrated in Supplementary Fig. S2), one can read off whether there is a symmetry breaking instability and which eigenmode (spherical harmonic mode in the case of a spherical cell) grows fastest. Experimental estimates exist only for a few of the parameters (see Supplementary Table S2). We therefore use linear stability analysis (which can be performed rapidly on a computer) to sample large numbers of parameter sets ($5 \times 10^6$) and identify those that are compatible with experimental observations (summarized in Supplementary Table S3). The resulting parameter sets span multiple orders of magnitude on each parameter axis (see Supplementary Fig. S3 and Supplementary Fig. S4), indicating that the model sloppy[76,77]. From these parameter sets, we pick a representative one (the one closest to the mean of the log parameters, see Supplementary Table S4) to generate the stability diagrams shown in Figs. 2, 4, and 5. To illustrate spontaneous polarization from a slightly perturbed homogeneous steady state as well as polarization induced by a local stimulus, we performed numerical simulations using COMSOL Multiphysics 5.4 (see Supplementary Movies 1–6).

## Experiments

**Media and strains.** All used media has the same base with 0.69% w/v Yeast nitrogen base (Sigma) +0.32% Amino acid mix (4× CSM) (Formedium) +2% Raffinose (Sigma). We used different galactose concentrations, denoted as $x$-Gal, where $x$ denotes the w/v % galactose percentage in the media.

An overview of all yeast strains used in this work is given in Table 2. Haploid strains yWKD065, yWKD069, YWKD070, yWKD071 and yWKD073 all originated from sporulation of diploids yWKD054

**Table 2 | Strains used in this work**

| Name | Genotype | Source |
|------|----------|--------|
| yLL3a | MATα can1-100, leu2-3, 112, his3-11,15, ura3Δ, BUD4-S288C | Ref. [3] |
| yLL112 | MATa/α CAN1/ can1::P_{mfa}-HIS3, leu2-3,112/leu2-3, 112, his3-11,15/his3-11,15, ura3Δ/ura3Δ, BUD4-S288C/BUD4-S288C, BEM1/ bem1:: KanMX6, BEM3/ bem3::NATMX4 | Ref. [3] |
| yWKD054b | MATa/α CAN1/ can1::P_{mfa}-HIS3, leu2-3,112/leu2-3, 112, his3-11,15/his3-11,15, ura3Δ/ura3Δ, BUD4-S288C/BUD4-S288C, BEM1/ bem1:: KanMX6, BEM3/ bem3::NATMX4, CDC42/ CDC42::URA3-P_{gal}-CDC42 | This work |
| yWKD055c | MATa/α CAN1/ can1::P_{mfa}-HIS3, leu2-3,112/leu2-3, 112, his3-11,15/his3-11,15, ura3Δ/ura3Δ, BUD4-S288C/BUD4-S288C, BEM1/ bem1:: KanMX6, BEM3/ bem3::NATMX4, CDC42/ CDC42::URA3-P_{gal}- sfGFP-Cdc42^{SW} | This work |
| yWKD065a | MATa, CDC42::URA3-P_{gal}-sfGFP-Cdc42^{SW}, can1::P_{mfa}-HIS3, leu2-3, 112, his3-11,15, ura3Δ, BUD4-S288C | This work |
| yWKD069a | MATa, bem1:: KanMX6, CDC42::URA3-P_{gal}-Cdc42-sfGFP^{SW}, can1::P_{mfa}-HIS3, leu2-3, 112, his3-11,15, ura3Δ, BUD4-S288C | This work |
| yWKD070a | MATa, bem1:: KanMX6, bem3::NATMX4, CDC42::URA3- P_{gal}-Cdc42-sfGFP^{SW}, can1:: P_{mfa}-HIS3, leu2-3, 112, his3-11,15, ura3Δ, BUD4-S288C | This work |
| yWKD071a | MATa, CDC42::URA3-P_{gal}-CDC42, can1:: P_{mfa}-HIS3, leu2-3, 112, his3-11,15, ura3Δ, BUD4-S288C | This work |

and YWKD055, using the lifted histidine auxotrophy for a-type haploids. Diploids yWKD054 and yWKD055 were generated by integration of plasmids pWKD010 and pWKD011 into yLL112[3] respectively. Plasmids pWKD010 and pWKD011 consist of a pRL368 backbone[78], with a URA3 selectable marker. After amplifying this backbone without GFP, homology regions upstream and downstream of endogenous CDC42 were added with Gibson assembly, separated by an EcoRI cut site. After cutting these plasmids with EcoRI (New England Biolabs), the homology flanks ensured the genomic integration during transformation replacing Cdc42 at its endogenous locus. Additionally, a superfolder GFP (sfGFP,[79], amino acid sequence GenBank: QLY89013.1) was added in pWKD011 with Gibson assembly between positions L134 and R135 of CDC42. This is based on previous work on *S. cerevisiae*, where a mCherry was integrated within Cdc42[13]. We eliminated the fitness effects from *mcherry-Cdc42^{SW}* by using a superfolder GFP protein, as suggested by work in *S. pombe* (Bendezú et al., 2015). Plasmids and genomic integrations were verified by sequencing.

The assays presented in Fig. 3 did not necessitate sfGFP, as planned localization experiments using fluorescence microscopy suffered from incomplete sfGFP degradation as documented previously in literature[80]. We tested whether the results presented, such as the growth rate differences across galactose levels, are not an artifact of adding this fluorophore, or auxotrophy differences across strains. We confirmed that the presence of the sfGFP insertion did not affect the growth rate of cells with CDC42 under the Gal promoter significantly for various galactose conditions (see Supplementary Fig. S5 and Supplementary Tables S6 and S7). Moreover, medium was supplemented with four times the normal amino acid concentrations to address differences in auxotrophies between yLL3a and the other strains, and no difference was observed in maximum growth rates of YWKD065a and yLL3a in Fig. 3.

**Growth rate assays.** We used a plate reader (Infinite M-200 pro, Tecan) for growth rate assays, with 96 well plates from Thermo Scientific, Nunc edge 2 96 F CL, Nontreated SI lid, CAT.NO.: 267427. Rows A and H and the columns 1 and 12 were not used for measurements. We inoculated a 96-well plate with 100 µl of medium and 5 µl of cells (from glycerol stocks) in each well, and grew the cells in 96-well plate for 48 h at 30 °C in a warm room. Afterwards the cells were diluted 200× into a new 96 well plate, which were then placed in the plate reader and the OD600 was measured for 48 h using a combination of linear and orbital shaking at 36 °C. We used a home-written data analysis program in Matlab[81] to determine the log-phase doubling time for every well. The doubling time was approximated by fitting the slope of the linear regime of the log plot of the raw data. We performed at least two different experiments per condition, and we performed at least 4 technical replicates per strain-condition combination (except at 2% galactose); see Supplementary Table S5.

The posteriors of non-WT backgrounds followed from normalization to WT rates by Monte Carlo simulations of the quotient of the original, non-normalized growth rate posteriors in a genetic background and the WT posterior in that medium. The non-normalized posteriors were calculated using the Metropolis-Hastings algorithm[82], from a rectangular prior and Student-t likelihood functions of doubling time fit estimates of all replicates in that medium. The standard errors of individual estimates come from the standard error of the slope parameter resulting from weighted least squares (WLS) on a moving window per OD curve, using an instrument error proxy for the WLS weights. The standard errors of individual estimates are corrected for overdispersion by the average modified Birge ratio[83] across media for WT.

**Microscopy assays**
**Cell imaging during growth rate assays.** The microscopy images were taken with a Nikon Eclipse Ti-E inverted microscope with an oil immersion 60× objective, with NA 1.40 and zoom factor 1.5. We used a 96 black multiwell plates compliant to the SBS (Society for Biomolecular Screening) standard-format with cover glass bottoms made from borosilicate glass.

Cells were incubated using the first part of the growth rate assay protocol (in the plate reader at 30 °C for 48 h). Then, they were diluted 100× to a new plate and incubate at 36 °C for 24 h, before they reached complete saturation. The cells were diluted 1000× for the WT backgrounds for all galactose concentrations, and 100× for non-WT backgrounds for galactose concentrations greater than 0.05%.

The media used for incubating and diluting the cells was 4xCSM +2% Raffinose with the respective galactose concentrations, for each strain.

**Cell size quantification.** All microscopy images were taken with an Olympus IX81 inverted microscope equipped with Andor revolution and Yokogawa CSU X1 modules. We used a 100× oil objective. The acquisition software installed is Andor iQ3. The CG imaging plates were from Zell-Kontakt. They are black multiwell plates compliant to the SBS (Society for Biomolecular Screening) standard-format with cover glass bottoms made from borosilicate glass.

Cells were grown in an overnight culture in CSM+ 2% Raffinose +2% Galactose media, without reaching saturation. On the next day, three washing steps with CSM+ 2% Raffinose were performed and subsequently the cells were re-suspended in the desired media of 0%, 0.06% and 0.1% Galactose. To obtain cell populations at all galactose concentrations, we first incubated all strains in 2% galactose concentration, where Cdc42 is highly overexpressed, such that also *bem1Δ* cells are able to efficiently polarize. After 15 h of incubation in 2% galactose concentration, we exchanged the medium to the desired galactose concentration. After 24 h, we observed the cells with light microscopy. After 24 h leftover Cdc42 from the initial 2% galactose

concentration incubation is (very low due to degradation and dilution (Cdc42 half-life is about 8 h[84]). From these images, we determined the average cell radius of the cells in the population.

Note that all of them contain the same base media: CSM+ 2% Raffinose. Afterwards the cells were incubated for 8 h at 30 °C followed by an imaging session, and subsequently incubated for another 16 h after which another imaging sessions was performed. We performed three independent experiments for each galactose concentration.

**Microscopy data analysis.** We performed bright field microscopy assays to monitor the cell size across different levels of Cdc42 in different genetic backgrounds. With ImageJ we manually determined the perimeter of the individual cells by fitting the live cells to a circle with the Measure tool. We performed three independent experiments per condition and per strain. In addition, we visually checked how many of the cells were alive and how many were dead based on their morphology. We observe what is called accidental cell death[85] upon inducing a very low Cdc42 protein copy number regulated by the Gal promoter. This type of cell deaths shows very distinctive phenotype associated to necrosis, namely: disintegration of cell structure and plasma membrane rupture (see Supplementary Fig. S7). Once we observe this phenotype in our cells, we classify them as dead. The error bar on the fraction of dead cells as well as of the average cell radius, is calculated as the standard error over the total number of analyzed cells.

### Reporting summary
Further information on research design is available in the Nature Portfolio Reporting Summary linked to this article.

## Data availability
The experimental data generated in this study have been deposited in the 4TU repositories: raw microscopy [https://doi.org/10.4121/60bea990-b1a6-40c7-9355-584e061791d5.v2] and growth rates [https://doi.org/10.4121/67e56fe8-6b54-446b-aaac-cd2e245ee066.v1]. Parameter sets for the mathematical model are available in GitHub [https://github.com/f-brauns/yeast-polarity-LSA]. Source data are provided with this paper.

## Code availability
Mathematica code for linear stability analysis of the mathematical model and COMSOL Multiphysics simulation files are provided in GitHub [https://github.com/f-brauns/yeast-polarity-LSA]. This repository also contains parameter sets filtered from large scale parameter space sampling. The Matlab and Python scripts used to analyze experimental data are provided in the following Github repository [https://github.com/leilaicruz/Experimental-data-analysis-protein-copy-number-in-polarity].

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

## Acknowledgements
We thank Felix Meigel and Marit Smeets for their pioneering experimental work. We thank Eelco Tromer for his advice on phylogenetics. We thank Daniel Needleman and Andrew Goryachev for critical reading of the manuscript. EF acknowledges support from the Deutsche Forschungsgemeinschaft (German Research Foundation) through the Collaborative Research Center 1032—Project ID 201269156—and the Excellence Cluster ORIGINS under Germany's Excellence Strategy—EXC-2094—390783311. L.L. and L.I.C. acknowledge support from the Netherlands Organization for Scientific Research (NWO) through a VIDI grant (016.Vidi.171.060). L.L. and W.D. acknowledge support from the Netherlands Organization for Scientific Research (NWO/OCW), as part of the Gravitation Program: Frontiers of Nanoscience. L.L. acknowledges funding from the European Research Council (ERC) under the European Union's Horizon 2020 research and innovation program (*Grant agreement No. 758132*).

## Author contributions
F.B., L.I.C., W.D., J.H., L.L., and E.F. designed research; F.B., J.H., and E.F. designed the theoretical models and performed the mathematical analyses; L.I.C., W.D., and I.B. and L.L. designed and carried out the experiments; F.B., L.I.C., W.D., L.L. and E.F. wrote the paper.

## Funding

## Competing interests
The authors declare no competing interests.
