## [Peer Review File · Nature Communications]

REVIEWER COMMENTS

Reviewer #1 (Remarks to the Author):

In this work, Brauns et al. use mathematical modeling as well as experiments to study the mechanism of Cdc42-mediated cell polarization in budding yeast. This yeast has been used extensively as a model organism to understand this important biological process. The authors put a great effort to consider evidence from prior experimental as well as theoretical work in order to propose a model that extends prior models for Cdc42 polarization. The three proposed redundant mechanisms in this model (Cdc42 transport, GEF transport, and GAP saturation) provided predictions that were tested experimentally, and are indicative of the adaptation and evolution of the polarization machinery. Overall, this is a detailed work that advances the field. However, I have some major as well as a few minor comments.

Major:

1. The redundancy between actin-based Cdc42 and GEF transport has been discussed in related contexts extensively before. My understanding is that a main novel aspect of this work is the additional “GAP saturation” mechanism that provides a Cdc42-concentration-dependent latent polarization mechanism in cells lacking Bem1 or both Bem1 and Bem3. This is shown by explicitly adding GAP and GAP-Cdc42 complexes to prior models of polarization. Better highlighting of this new feature might strengthen the novelty of the paper which can otherwise come across as incremental or speculative (the discussion on adaptation and evolution is inspiring but not as much supported by data as the title implies, for example).

2. Given my evaluation in comment 1, the paper would benefit by a more in-depth analysis and/or discussion of the GAP saturation mechanism. For example, in the model GAPs are assumed to be bound membrane. Is this aspect of the model supported experimentally and is it essential? What is known about GAP localization in budding yeast and can the GAP saturation mechanism be tested experimentally? How does this model compare to prior modeling efforts that explicitly accounted for GAPs? (the works that come to mind by Howell et al. <https://doi.org/10.1016/j.cell.2012.03.012>, Khalili et al. <https://doi.org/10.1371/journal.pcbi.1006317> and Okada et al. <https://doi.org/10.1016/j.devcel.2013.06.015>)

3. I found it somewhat hard to get a good sense of the model, which is central to this work, without resorting to Supplementary Information. I believe that moving the main assumptions and equations of the model, such as section 2.2 and parts of section 1 of the Supplementary Information, into the Materials and Methods, would help readers like me (here I assume that Materials and Methods will appear in the same published document as the main text).

4. The experimental data in Fig. 3 seem noisy, especially at low values of galactose for Δ_{bem1} . The error bars also seem to be wide for 68% credibility (is this 68% probability that the average is in this window?). Additionally, at some galactose concentrations the value is zero for the fitness of Δ_{bem1} , Δ_{bem3} while at lower values it had a finite value; were none of the cells growing at 0.05 galactose concentration?

5. When the model is compared to experiments with Cdc42-ritC fusion chimeras, these chimeras are discussed as “permanently membrane-bound” and the dissociation rate of Cdc42 is set to 0. However, the FRAP experiments in Kang et al. show finite recovery (indicating a non-zero dissociation rate), even if slower than that of GFP-Cdc42. Can the authors address this apparent discrepancy?

Minor Comments:

- Is the cytosol concentration c_G in Eq. (6b) and (9b) supposed to be zero?
- Page 2, line 56 missing the word “a” in “develop a mathematical”
- Page 4, line 159; Page 5, line 186, and a few other places there are unformatted references
- Page 5, line 188 there is an extra parenthesis
- Page 7, line 312 this should be Figure 4H, not Figure 3H
- Figure 4A, caption, i-iii do not seem to correspond to any label on the image

Reviewer #2 (Remarks to the Author):

A. Summary of the advance made in this paper and its potential significance to the field.

In this manuscript, Brauns et al. propose a theoretical framework for the experimental data obtained in [1] that shows how budding yeast cells recover from a critical perturbation, particularly the deletion of BEM1, through the deletion of two Cdc42 GAPs, Bem2 and Bem3. Bem1 is a protein essential for the positive feedback loop that enables Cdc42 polarization and spontaneous symmetry breaking. Bem1 directly binds both the PAK Cla4 and the GEF Cdc24 and Cdc42-GTP during the polarization of Cdc42 that occurs during late G1. The mathematical framework of this manuscript is based on the one proposed in a previous work [2], to which they have added three incremental extensions based on past experimental data. Considering the extensive theoretical and empirical work that has been done depicting polarization in budding yeast during the last decades, the predictions shown in this work might seem

rather logical and previously demonstrated in different backgrounds in other works. For instance, that cells have backup systems (or rescue mechanisms) to break symmetry was suggested in [3], showing that a hyperactive *cdc42* mutant allele can live without Cdc24. This experiment is conceptually similar to what they have theoretically modeled in the simulation depicted in video 3, in which increasing the dosage of Cdc42 allows Cdc42 clustering in *bem1Δ* cells. The data obtained in [3] suggest that any manipulation that results in increased Cdc42 activity (such as GAP inactivation or deletion or an increase in Cdc42 levels) might allow the cells to polarize in the absence of Bem1, which partially impairs Cdc24 local recruitment and activation. In addition, a mathematical model of converging linear feedback via GEF and GAP for symmetry breaking has already been suggested in the literature [4]. Although the work seems to have been performed very competently and the hypothesis proposed about Bem1 evolution in the conclusion and discussion is appealing and biologically relevant, the current paper's scientific data do not add any significant advance in the mechanisms involved in the robustness of polarity establishment in budding yeast.

B. Major points for improvement of the manuscript:

1. Authors indicate that one of the extensions to the model previously published in [2] is to account for vesicle-based transport of Cdc42 towards zones of high Cdc42-GTP concentration. However, some authors have questioned actin-mediated vesicle traffic's significant role in polarizing Cdc42 [5, 6]. Many studies performed in budding and fission yeast have suggested that actin-based vesicle trafficking plays a vital role in polarization. Still, the key cargo is unlikely to be Cdc42. Therefore, the authors might try to validate their model without considering vesicle-based transport of Cdc42 towards zones of high Cdc42-GTP concentration.
2. It is not clear whether the authors have included the negative feedback described within the polarity network in budding yeast, which is essential for enhancing robustness in the response [7]. For instance, in [7], it is shown that overexpression of Cdc42 in the simulations causes the spreading of GTP-Cdc42 uniformly. In contrast, a polarized steady state is generated when negative feedback is taken into consideration. Therefore, negative feedback makes polarization more robust in fluctuating concentrations of polarity factors. It is also striking why the authors have not cited this seminal paper. This should be revisited.
3. The authors have not considered in their model the positive feedback exerted by Rsr1, and the justification they offer is poorly justified if the theoretical model they proposed is focused on recapitulating all the polarity networks that contribute to polarized growth. In mutants lacking transmembrane landmark proteins which bud at random locations but express Rsr1, the Bem1-dependent positive feedback is not essential for polarization [8]. One conclusion from these experiments is that Rsr1 must also contribute to the spontaneous, landmark-independent polarization of Cdc42. Like Bem1, Rsr1-GTP binds the Cdc42 GEF Cdc24 and promotes its release from autoinhibition.

4. Growth experiments shown in Figure 3 should be accompanied by micrographs showing cell morphology and budding at different galactose concentrations.
5. Line 288. The prediction of the rescue mechanisms of longer residence times of Cdc42 at lower GAP activity should be tested experimentally.
6. Lines 298-309. This paragraph is not appropriate in the results section.
7. In Figure S6 the authors should show the percentage of budded and multinucleated cells. In addition, the level of Cdc42 should be shown by western blot.
8. The authors should include snapshots of the simulation videos in the corresponding figure in the paper.

C. Minor points

Line 159- Error. Reference source not found.

Line 186- Error. Reference source not found.

Supplementary information. Page 23. The authors should replace Cdc42 for Cdc24 in the sentence following references 32 and 60. Later in the paragraph, the sentence preceding reference 49 is not complete. It should end with Snc2.

REFERENCES

1. Laan, L., J.H. Koschwanetz, and A.W. Murray, Evolutionary adaptation after crippling cell polarization follows reproducible trajectories. *eLife*, 2015. 4.
2. Klünder, B., et al., GDI-Mediated Cell Polarization in Yeast Provides Precise Spatial and Temporal Control of Cdc42 Signaling. *PLoS Computational Biology*, 2013. 9(12): p. e1003396.

3. Caviston, J.P., S.E. Tcheperegine, and E. Bi, Singularity in budding: a role for the evolutionarily conserved small GTPase Cdc42p. *Proc Natl Acad Sci U S A*, 2002. 99(19): p. 12185-90.
4. Goryachev, A.B. and M. Leda, Many roads to symmetry breaking: molecular mechanisms and theoretical models of yeast cell polarity. *Molecular Biology of the Cell*, 2017. 28(3): p. 370-380.
5. Woods, B. and D.J. Lew, Polarity establishment by Cdc42: Key roles for positive feedback and differential mobility. *Small GTPases*, 2019. 10(2): p. 130-137.
6. Martin, S.G., Spontaneous cell polarization: Feedback control of Cdc42 GTPase breaks cellular symmetry. *Bioessays*, 2015. 37(11): p. 1193-201.
7. Howell, Audrey S., et al., Negative Feedback Enhances Robustness in the Yeast Polarity Establishment Circuit. *Cell*, 2012. 149(2): p. 322-333.
8. Smith, S.E., et al., Independence of symmetry breaking on Bem1-mediated autocatalytic activation of Cdc42. *The Journal of cell biology*, 2013. 202(7): p. 1091-1106.

Reviewer #3 (Remarks to the Author):

Summary: The paper 'Adaptability and evolution of the cell polarization machinery in budding yeast' by Brauns et.al. presents a, to my knowledge, novel mathematical model of the important yeast polarisation machinery. This machinery has already been modelled by many groups, including by the authors of this manuscript, and it has emerged as one paradigm for pattern formation. The present paper presents several exciting new directions in this field. Most notably, the authors investigate how polarization can be achieved through different submodules of molecules. The authors demonstrate that polarisation can be robustly achieved even when some of these submodules are deleted.

I list several specific questions and comments below. If the authors can clarify these points, then I can recommend the paper for publication.

Specific comments or questions:

The model description is ok, but it may be hard to reproduce all results exactly from the description. By far the best way to ensure full reproducibility is to simply make all the code available, e.g. on Github. I have seen some references to supplementary Mathematica files which is great; however, I cannot find the files anywhere. Apologies if I overlooked this. For the Comsol simulations, I cannot even find a link anywhere.

If the initial conditions were purely homogenous with a small (and spatially uniformly?) noise term, why does the polarisation cap always appear in the upper right part of the cell?

The parameter investigation seems to indicate that the system is sloppy, but that was not formally shown. The authors estimate a mean parameter set (Fig S3). But it appears that this a lot of parameter combinations are compatible with the experimental constraints. The key question is if the major results of the paper are robust with respect to choices of parameters that are compatible with the constraints?

Comparison of theory with experiments: The paper is a great theory paper, and the authors clearly demonstrate compatibility of their results with published experiments. The authors also show a few new experimental results on cell growth and fitness under various conditions. However, there seems to be no direct new experiment that confirms the simulations (e.g. fluorescent imaging of the molecules involved in the authors model). To me, this is perfectly fine, as the paper is already very strong with the new model as presented. However, I suggest the authors rephrase sentences such as 'Our theory, confirmed by experimental analysis, reveals...' to 'Our theory, which is compatible with published experiments ...'. As the authors indicate themselves, many new experiments should be performed to validate the predictions (and therefore, fully validate the model).

Title: I suggest a title that does not focus on adaptability or evolution. I really like Figure 5 and the text in the Conclusions and Discussions section, suggesting an evolutionary mechanism. However, I suppose the authors placed in into this section (and not into Results) exactly for the reason that this is a suggestion of how things could have evolved. There is no strong confirmation for this yet. Hence, I would choose a title that is more representative of the results for which the authors obtain strong support, e.g. robustness and the different submodules, rather than adaptability and evolution.

Language: Is protein dosage the best word? I would simply say: protein copy numbers, as this is apparently what the authors refer to?

Language: Occasionally sentences are very long and hard to digest. Example: 'This significant difference in Cdc42 expression level sensitivity between the WT mechanism and the rescue mechanism is in the context of our theory explained by the qualitative difference of their principles of operation, as we discussed above in the The Cdc42 interaction network facilitates a latent polarization mechanism'.
'Suggestion: break such long sentences down into 2-3 parts.

L5: the polarity machine is robust to genetic perturbations: is this statement true in generality? I am not a yeast expert, could the authors make sure that this sentence is precise? E.g., is there robustness with respect to the deletion of any (known) individual gene of the polarity machinery? PS: after reading the whole paper it seems very misleading – the authors talk about robustness with respect to a single gene

L35/L426: 'Active Cdc42 directs both cytosolic diffusion: Diffusion itself is obviously random movement and not 'directed'. Of course, as the authors explain, they mean diffusion driven by concentration gradients. However, diffusion inevitably requires a gradient; hence I think the word 'directed' is misleading in the present context. I suggest removing 'directed' when talking about diffusion, and just using 'directed' in the context of transport along actin cables

L43.. Clearly state that Cdc24 is the GEF

L166: 'As expected, WT cells grow at all galactose concentrations.' . If I understand it correctly, Fig. 3 is normalised to wild type. Hence, by definition, WT has a fitness of 1. So how can we see that this statement is true? Ok, if the fitness is rescaled to 1, we know it is not zero, but it could be very small. It may be good to show the not-rescaled values somewhere in the SI (Table S6 seems to show only the fraction of wells with proliferation. Or is that how the authors define the fitness? Is a precise definition of fitness given anywhere?)

L56 typo: 'a' missing

L159, 186 and maybe elsewhere: reference was broken. The authors should check thoroughly the references before the resubmission

L353: typo: =

SI p3: The authors cite SI reference [22] for their argument that dissociation of the Cdc42 complex with a GAP is rate-limiting. First, do they mean rate-limiting compared to any other rates in the model (e.g. transport rates)? Second, if the dissociation is slow, is a Michaelis-Menten approximation justified, which was used in [22]? Also, [22] seems to deal with human proteins in vitro – are these results transferable to yeast (the main text line 114 seems to indicate the reference is about yeast, but glancing at the paper it does not appear to be so). Finally, the determined parameter k_{gt} in Table S5 seems to be of the same order of magnitude as the other rates (or even larger in the scenario for Fig. 4)

SI p3: The authors argue that Cdc24 is distributed uniformly on the membrane. I do not fully understand the argument. If Cdc24 can interact with Cdc42 which is non-uniform on the membrane, then Cdc24 should also possibly become non-uniform on the membrane due to these interactions. Lower GEF activity does not seem necessary to imply this. It may depend on what exactly lower GEF activity means. If the binding of Cdc24 to 42 is the same but the subsequent rate of Cdc42 activation (e.g. GTP binding)

is lower, I am not sure if that would imply that there is less of a non-uniformity of Cdc24 than with full GEF activity. Moreover, a 50% reduction in activity is not that high. It would be better if the authors could quickly check their assumptions, which should not be too hard.

SI p6: The authors argue about the similarities between vesicle-based and diffusion-based transport. However, I would expect vesicle based transport along the cytoskeleton to lead to an advection instead of a diffusion term.

SI p8/9: It appears there are 7 membrane bound species, yet the vector containing all membrane reactions, eq (5), has only 6 elements. Did I miss anything?

SI p15, step I. Typo $0:15 \times 10^{-2}$?

SI p18, Step III: Why is $k_d = 0$ modelling the lack of recycling via the cytosol? Could one not argue no recycling corresponds to the attachment rate being zero, as opposed to the detachment rate?

SI p18, bottom: 'that is unlikely' -> it missing

SI, p18: 'to same small value' -> the same

SI p22, bottom. Are the authors really referring to Video 6 here? Also, while it appears that the pattern is indeed stable after removal of the stimulus, have the authors explicitly shown this, at least through much longer simulations to confirm nothing changes?

Below, comments by the reviewers are in *blue italic* text, our responses are in black.

Reviewer #1

In this work, Brauns et al. use mathematical modeling as well as experiments to study the mechanism of Cdc42-mediate cell polarization in budding yeast. This yeast has been used extensively as a model organism to understand this important biological process. The authors put a great effort to consider evidence from prior experimental as well as theoretical work in order to propose a model that extends prior models for Cdc42 polarization. The three proposed redundant mechanisms in this model (Cdc42 transport, GEF transport, and GAP saturation) provided predictions that were tested experimentally, and are indicative of the adaptation and evolution of the polarization machinery. Overall, this is a detailed work that advances the field. However, I have some major as well as a few minor comments.

Major:

1. The redundancy between actin-based Cdc42 and GEF transport has been discussed in related contexts extensively before. My understanding is that a main novel aspect of this work is the additional “GAP saturation” mechanism that provides a Cdc42-concentration-dependent latent polarization mechanism in cells lacking Bem1 or both Bem1 and Bem3. This is shown by explicitly adding GAP and GAP-Cdc42 complexes to prior models of polarization. Better highlighting of this new feature might strengthen the novelty of the paper which can otherwise come across as incremental or speculative (the discussion on adaptation and evolution is inspiring but not as much supported by data as the title implies, for example).

We thank the referee for the critical reading and positive assessment of our work and for suggesting how we can strengthen the novelty of the paper. We have added a sentence in the introduction that emphasizes this novel aspect of our model and its importance for the results.

2. Given my evaluation in comment 1, the paper would benefit by a more in-depth analysis and/or discussion of the GAP saturation mechanism. For example, in the model GAPs are assumed to be bound membrane. Is this aspect of the model supported experimentally and is it essential? What is known about GAP localization in budding yeast and can the GAP saturation mechanism be tested experimentally? How does this model compare to prior modeling efforts that explicitly accounted for GAPs? (the works that come to mind by Howell et al.

<https://doi.org/10.1016/j.cell.2012.03.012>, Khalili et al.

<https://doi.org/10.1371/journal.pcbj.1006317> and Okada et al.

<https://doi.org/10.1016/j.devcel.2013.06.015>)

The referee raises several important questions here. Let us start by answering the last question: None of these prior models have accounted for the intermediate Cdc42-GAP complex and can therefore not exhibit the enzyme-saturation effect that is at the core of the rescue mechanism we propose. These models therefore cannot describe polarization in the absence of Bem1. Instead, these studies are focussed on more

fine-grained quantitative details (such as the size of the polar zone) of the Bem1-dependent wild-type polarization machinery.

Enzyme saturation is a generic property of enzymatic kinetics. It is particularly strong if the catalysis and dissociation step is slow compared to the enzyme-substrate binding. This has been shown for the GAPs of human Cdc42 (see ref [22] in the SI). One potential experimental test for GAP saturation, as also suggested by referee 2, is to measure the residence time of Cdc42 on the membrane, e.g. via FRAP on the different regions. GAP saturation would imply that the residence time in the polar zone is much longer than elsewhere on the membrane. Unfortunately, it is hard to execute this experiment in a quantitatively convincing manner because of the size of budding yeast. Typical FRAP areas comprise a large part of the yeast cell making it difficult to quantitatively disentangle the polar zone from the rest of the membrane.

Finally, the question of GAP localization. In the model we assume that the GAPs are membrane bound. However, strict membrane binding is not crucial for the model. The results are qualitatively unchanged when the free GAPs can cycle between membrane and cytosol. GAP saturation can drive Cdc42 polarization as long as resupply of GAPs in the incipient polar zone is not faster than accumulation of Cdc42. This resupply could happen via diffusion of the cytosolic fraction of GAPs. Experiments show that only about 10% of Bem3 is cytosolic and that Bem3 membrane binding is mediated by a PH domain which binds directly to phospholipids in the membrane. While Rga1/2 are not known to have a PH domain, they localize to the bud tip and bud neck during cell polarization and budding [Caviston et al. *MBoC* 14, 4051–4066 (2003)]. We therefore assume that a significant fraction of them is bound to the membrane (or cortex) during polarity establishment such that a fast resupply of free GAPs in the incipient bud-site which would relieve GAP saturation there is unlikely. Moreover, since GAPs are significantly larger than Cdc42 (ca 10x larger molecular weight), their diffusion will be significantly slower than that of Cdc42. Thus, even if there is a significant pool of cytosolic GAPs, their diffusion would be too slow to relieve GAP saturation. These qualitative statements could be quantified by extending the model to account for GAP attachment and detachment at the membrane. However, this would be a distraction from the core topic of redundant functional submodules. We therefore leave such model extensions for future work.

3. I found it somewhat hard to get a good sense of the model, which is central to this work, without resorting to Supplementary Information. I believe that moving the main assumptions and equations of the model, such as section 2.2 and parts of section 1 of the Supplementary Information, into the Materials and Methods, would help readers like me (here I assume that Materials and Methods will appear in the same published document as the main text).

We thank the referee for this excellent suggestion. We have moved the central model description and the mathematical equations into the Methods section. The more in-depth discussion of specific model aspects is retained in the SI.

4. The experimental data in Fig. 3 seem noisy, especially at low values of galactose for Δ_{bem1} . The error bars also seem to be wide for 68% credibility (is this 68%

*probability that the average is in this window?). Additionally, at some galactose concentrations the value is zero for the fitness of *del_bem1*, *bem3* while at lower values it had a finite value; were none of the cells growing at 0.05 galactose concentration?*

Indeed the 68% probability the average is inside the window. In some data points, e.g., the Gal1-sfGFP-Cdc42 BEM1 BEM3, the large interval originates from the technical replicate variation, i.e., noise across runs/wells. However, for the *dbem1* at low galactose concentrations, there is very infrequent growth, e.g., at 0.03% galactose there is actually growth only once. In this case, the variance of only one student t likelihood, which depends on the fitting error corrected for overdispersion, is the main determinant for the width of the credible interval, so this makes this interval a bit wide. However, the poor fitness of this background will likely still yield large credible intervals even with a large number of replicates, as its low fitness also makes it vulnerable for suppressor sweeps of the population. The stochasticity in successfully growing up wells in low galactose concentrations for *dbem1* backgrounds is also seen in the other points. Indeed, at 0.04% and 0.05%, no growth is exhibited in 4 and 6 tries each of *dbem1dbem3* and *dbem1* respectively, while there is a single run out of 6 for the *dbem1* that grew at 0.03%.

5. When the model is compared to experiments with Cdc42-ritC fusion chimeras, these chimeras are discussed as “permanently membrane-bound” and the dissociation rate of Cdc42 is set to 0. However, the FRAP experiments in Kang et al. show finite recovery (indicating a non-zero dissociation rate), even if slower than that of GFP-Cdc42. Can the authors address this apparent discrepancy?

Setting the dissociation rate to 0 is a simplifying assumption to show that the model is robust even in the extreme case of no dissociation. The results are qualitatively unchanged if one uses some small but non-zero dissociation rate. In fact, weak dissociation will always enhance the pattern formation instability because it allows a fraction of Cdc42 to diffuse in the cytosol which accelerates accumulation in the polar zone. We have added a remark addressing his point in SI Sec. 4 (Step III).

Minor Comments:

• Is the cytosol concentration c_G in Eq. (6b) and (9b) supposed to be zero?

Yes, we have removed them for clarity.

• Page 2, line 56 missing the word “a” in “develop a mathematical”

• Page 4, line 159; Page 5, line 186, and a few other places there are unformatted references

• Page 5, line 188 there is an extra parenthesis

• Page 7, line 312 this should be Figure 4H, not Figure 3H

We thank the referee for pointing out these typos and have fixed them in the manuscript.

• Figure 4A, caption, i-iii do not seem to correspond to any label on the image.

We thank the referee for pointing us to this mistake. We have rewritten the caption to match the figure content.

Reviewer #2

A. Summary of the advance made in this paper and its potential significance to the field.

In this manuscript, Brauns et al. propose a theoretical framework for the experimental data obtained in [1] that shows how budding yeast cells recover from a critical perturbation, particularly the deletion of BEM1, through the deletion of two Cdc42 GAPs, Bem2 and Bem3. Bem1 is a protein essential for the positive feedback loop that enables Cdc42 polarization and spontaneous symmetry breaking. Bem1 directly binds both the PAK Cla4 and the GEF Cdc24 and Cdc42-GTP during the polarization of Cdc42 that occurs during late G1. The mathematical framework of this manuscript is based on the one proposed in a previous work [2], to which they have added three incremental extensions based on past experimental data. Considering the extensive theoretical and empirical work that has been done depicting polarization in budding yeast during the last decades, the predictions shown in this work might seem rather logical and previously demonstrated in different backgrounds in other works. For instance, that cells have backup systems (or rescue mechanisms) to break symmetry was suggested in [3], showing that a hyperactive cdc42 mutant allele can live without Cdc24. This experiment is conceptually similar to what they have theoretically modeled in the simulation depicted in video 3, in which increasing the dosage of Cdc42 allows Cdc42 clustering in bem1Δ cells. The data obtained in [3] suggest that any manipulation that results in increased Cdc42 activity (such as GAP inactivation or deletion or an increase in Cdc42 levels) might allow the cells to polarize in the absence of Bem1, which partially impairs Cdc24 local recruitment and activation. In addition, a mathematical model of converging linear feedback via GEF and GAP for symmetry breaking has already been suggested in the literature [4]. Although the work seems to have been performed very competently and the hypothesis proposed about Bem1 evolution in the conclusion and discussion is appealing and biologically relevant, the current paper's scientific data do not add any significant advance in the mechanisms involved in the robustness of polarity establishment in budding yeast.

We thank the referee for carefully reading our manuscript and providing constructive feedback. To start with, we would like to emphasize that while our study is mainly theoretically driven, based on modeling and computational analysis, we test key theoretical predictions in experiments. This provides additional support for the model which explains a number of previously puzzling experimental observations in a unified framework. Moreover, our extensive computational analysis has revealed the importance of enzyme saturation for spatio-temporal dynamics and has allowed us to formulate a concrete evolutionary hypothesis. We think that these are valuable insights into the mechanisms involved in the robustness of polarity establishment in budding yeast.

We hope that our point-by-point replies below address all the referees' concerns.

B. Major points for improvement of the manuscript:

1. Authors indicate that one of the extensions to the model previously published in [2] is to account for vesicle-based transport of Cdc42 towards zones of high Cdc42-GTP concentration. However, some authors have questioned actin-mediated vesicle traffic's significant role in polarizing Cdc42 [5, 6]. Many studies performed in budding and fission yeast have suggested that actin-based vesicle trafficking plays a vital role in polarization. Still, the key cargo is unlikely to be Cdc42. Therefore, the authors might try to validate their model without considering vesicle-based transport of Cdc42 towards zones of high Cdc42-GTP concentration.

Vesicle-based transport is not a necessary ingredient in our model. In fact we model diffusive transport in the cytosol and vesicle-based transport as interchangeable on a coarse-grained level. This entails that vesicle-based transport is just one of multiple pathways that transport Cdc42 (and other proteins) towards the polar zone where Cdc42-GTP concentration is high.

2. It is not clear whether the authors have included the negative feedback described within the polarity network in budding yeast, which is essential for enhancing robustness in the response [7]. For instance, in [7], it is shown that overexpression of Cdc42 in the simulations causes the spreading of GTP-Cdc42 uniformly. In contrast, a polarized steady state is generated when negative feedback is taken into consideration. Therefore, negative feedback makes polarization more robust in fluctuating concentrations of polarity factors. It is also striking why the authors have not cited this seminal paper. This should be revisited.

We thank the referee for noting that we overlooked ref [7] which we cite. We have not included the negative feedback described in [7]. In fact, the phase diagrams in Fig. 2 show that both the WT as well as the rescue mechanism are fairly tolerant against Cdc42 overexpression, even without the negative feedback loop. Nonetheless it would be interesting to extend our model to include additional feedback mechanisms in the future. For the current work, our goal is not to model all feedback loops, which is likely infeasible due to the large number of interaction partners of Cdc42, but to study the redundancy in the core interaction network of Cdc42. Modeling more feedback loops might yield further redundant functional submodules.

We have now added a reference to [7] in the manuscript mentioning the robustness conveyed by the negative feedback loop.

3. The authors have not considered in their model the positive feedback exerted by Rsr1, and the justification they offer is poorly justified if the theoretical model they proposed is focused on recapitulating all the polarity networks that contribute to polarized growth. In mutants lacking transmembrane landmark proteins which bud at random locations but express Rsr1, the Bem1-dependent positive feedback is not essential for polarization [8]. One conclusion from these experiments is that Rsr1 must also contribute to the spontaneous, landmark-independent polarization of Cdc42. Like Bem1, Rsr1-GTP binds the Cdc42 GEF Cdc24 and promotes its release from autoinhibition.

In [8] (Smith et al., 2013), the viability of *bem1Δ* cells is similar to that of *bem1Δ rsr1Δ* cells. Thus, even if Rsr1 might provide a feedback loop for Cdc42-polarization, this does not translate into successful budding and proliferation. Moreover, *rsr1Δ* has no significant effect on viability. Together these findings suggest that Rsr1 supports only a weak feedback loop. In our model, we account for such weak feedback loops, by an effective recruitment of Cdc42 to the membrane by membrane-bound Cdc42-GTP (see Fig. 1E). Rsr1 might be one of many interaction partners of Cdc42-GTP that provides such a recruitment.

In principle, one could add additional feedback loops. For instance, one that enhances the GEF activity where Cdc42-GTP concentration is high. Adding such feedback loops would only add to the overall claim of the manuscript, namely that there is a high degree of redundancy in the Cdc42-polarization machinery. However, this would also further complicate the modeling and model analysis and we believe the manuscript is clearer without such additional complications.

We now mention Rsr1 as a potential feedback mediator in the caption of Fig. 1 and have added a passage in the SI that discusses the potential additional redundancy afforded by an Rsr1-mediated feedback loop.

4. Growth experiments shown in Figure 3 should be accompanied by micrographs showing cell morphology and budding at different galactose concentrations.

We added micrographs of the different mutants after 24 hours of incubation, for 0.06% galactose, where the difference in phenotype between the different mutants is the largest (see Fig. 3B).

5. Line 288. The prediction of the rescue mechanisms of longer residence times of Cdc42 at lower GAP activity should be tested experimentally.

We agree with the referee that this would be a very interesting experiment. One potential experimental test for GAP saturation is to measure the residence time of Cdc42 on the membrane, e.g. via FRAP of the different regions. GAP saturation would imply that the residence time in the polar zone is much longer than elsewhere on the membrane. Unfortunately, it is hard to execute this experiment in a quantitatively convincing manner because of the size of budding yeast. Typical FRAP areas comprise a large part of the yeast cell making it difficult to quantitatively disentangle the polar zone from the rest of the membrane. Therefore we consider this experiment outside of the scope of the current manuscript. We have reduced the weight we put on the experimental part by rephrasing some sentences, see our comment to referee 3.

6. Lines 298-309. This paragraph is not appropriate in the results section.

We have moved this paragraph about the parallels between the rescue mechanism in the Cdc42-network and Min-protein pattern formation in *E. coli* to the discussion.

7. In Figure S6 the authors should show the percentage of budded and multinucleated cells. In addition, the level of Cdc42 should be shown by western blot.

We added the percentage of multinucleated cells obtained using DAPI staining to Figure S6. We did not add the number of budded cells because the cells we analyzed in Figure S6 reached the stationary phase, and thus they are no longer budding. With our DAPI staining we observe some multinucleate cells, however this is a small fraction. Rather (as we showed already in Figure S6B) under slow growing conditions, we mostly observe dead cells. And with DAPI staining you cannot determine the number of nuclei in dead cells, because they become permeable and overstain. We know from previous work that *bem1Δ* cells explode, if they do not polarize (Laan et al 2015) and therefore we conclude that it is hard to detect multinucleated cells in the assay as we used it here.

We attempted to visualize the Cdc42 concentrations in stationary phase cells to assess if there was an accumulation of Cdc42 over the lifetime of the experiment. In Figure R1, we show the SDS page gels and anti-Cdc42 western blots from multiple strains under a variety of conditions. Although we see that the cell lysis was successful, it was not possible to visualize Cdc42 in the lysis through this method. We included a positive control in the form of a recombinantly purified Cdc42 protein from budding yeast. This showed a clear signal for Cdc42 in the western blot. We also attempted to visualize Cdc42 in log-phase cells as known Cdc42 concentrations were almost exclusively determined in log-phase cultures^{1,2,4-11}. Unfortunately, this was also unsuccessful.

Figure R1: A and C W303 cell lysates of cells grown in the presence of 0.01%, 0.02%, 0.05%, 0.06%, 0.08%, 0.1%, 0.2%, 2% Galactose (Lanes 1-8 respectively), cells isolated in their logarithmic growth phase in YPD (Lane 9), recombinantly purified yeast Cdc42 (Lane 10). **B and D** W303 cells with a *cdc42* placed behind a Gal

promotor grown in the presence of 0.01%, 0.02%, 0.05%, 0.06%, 0.08%, 0.1%, 0.2%, 2% Galactose (Lanes 1-8 respectively), purified Cdc42 (Lane 9). Approximately 1 µg of purified Cdc42 protein was loaded as a reference.

8. The authors should include snapshots of the simulation videos in the corresponding figure in the paper.

We have added snapshots from the simulations in Fig. 2.

C. Minor points

Line 159- Error. Reference source not found.

Line 186- Error. Reference source not found.

Supplementary information. Page 23. The authors should replace Cdc42 for Cdc24 in the sentence following references 32 and 60. Later in the paragraph, the sentence preceding reference 49 is not complete. It should end with Snc2.

We thank the referee for catching these typos.

REFERENCES

- 1. Laan, L., J.H. Koschwanez, and A.W. Murray, Evolutionary adaptation after crippling cell polarization follows reproducible trajectories. eLife, 2015. 4.*
- 2. Klünder, B., et al., GDI-Mediated Cell Polarization in Yeast Provides Precise Spatial and Temporal Control of Cdc42 Signaling. PLoS Computational Biology, 2013. 9(12): p. e1003396.*
- 3. Caviston, J.P., S.E. Tcheperegine, and E. Bi, Singularity in budding: a role for the evolutionarily conserved small GTPase Cdc42p. Proc Natl Acad Sci U S A, 2002. 99(19): p. 12185-90.*
- 4. Goryachev, A.B. and M. Leda, Many roads to symmetry breaking: molecular mechanisms and theoretical models of yeast cell polarity. Molecular Biology of the Cell, 2017. 28(3): p. 370-380.*
- 5. Woods, B. and D.J. Lew, Polarity establishment by Cdc42: Key roles for positive feedback and differential mobility. Small GTPases, 2019. 10(2): p. 130-137.*
- 6. Martin, S.G., Spontaneous cell polarization: Feedback control of Cdc42 GTPase breaks cellular symmetry. Bioessays, 2015. 37(11): p. 1193-201.*
- 7. Howell, Audrey S., et al., Negative Feedback Enhances Robustness in the Yeast Polarity Establishment Circuit. Cell, 2012. 149(2): p. 322-333.*
- 8. Smith, S.E., et al., Independence of symmetry breaking on Bem1-mediated autocatalytic activation of Cdc42. The Journal of cell biology, 2013. 202(7): p. 1091-1106.*

Reviewer #3

Summary: The paper 'Adaptability and evolution of the cell polarization machinery in budding yeast' by Brauns et.al. presents a, to my knowledge, novel mathematical model of the important yeast polarisation machinery. This machinery has already been modelled by many groups, including by the authors of this manuscript, and it has emerged as one paradigm for pattern formation. The present paper presents several exciting new directions in this field. Most notably, the authors investigate how polarization can be achieved through different submodules of molecules. The authors demonstrate that polarisation can be robustly achieved even when some of these submodules are deleted.

I list several specific questions and comments below. If the authors can clarify these points, then I can recommend the paper for publication.

We thank the referee for carefully reading our manuscript and providing . We hope our point by point replies resolve all his remaining concerns.

Specific comments or questions:

The model description is ok, but it may be hard to reproduce all results exactly from the description. By far the best way to ensure full reproducibility is to simply make all the code available, e.g. on Github. I have seen some references to supplementary Mathematica files which is great; however, I cannot find the files anywhere. Apologies if I overlooked this. For the Comsol simulations, I cannot even find a link anywhere.

All code, including the setup files for COMSOL simulations have been made available on Github [<https://github.com/f-brauns/yeast-polarity-LSA>] and are linked in the manuscript as well as the SI.

If the initial conditions were purely homogenous with a small (and spatially uniformly?) noise term, why does the polarisation cap always appear in the upper right part of the cell?

In simulations without the optogenetic cue, the position of the polar zone is random. For visualization of the simulations in Videos 1–6, the camera position was chosen after the simulation had finished in such a way that the polar zone appears in approximately the same position in each video. We have added this remark in the SI (Sec. 3).

The parameter investigation seems to indicate that the system is sloppy, but that was not formally shown. The authors estimate a mean parameter set (Fig S3). But it appears that this a lot of parameter combinations are compatible with the experimental constraints. The key question is if the major results of the paper are robust with respect to choices of parameters that are compatible with the constraints?

Indeed the system seems to have many sloppy parameters. Showing this formally is beyond the scope of this manuscript but is an interesting direction for future research. The parameter sets shown in the scatter plots in Fig. S4 and Fig. S5 were filtered to be compatible with the experimental observations. Since this includes the conditions where individual functional submodules of the polarization machinery are knocked out (by setting Bem1 density to zero and by making Cdc42 permanently membrane bound), all filtered parameter sets exhibit threefold redundancy of Cdc42 polarization mechanisms and are therefore compatible with the key claim of the paper.

Comparison of theory with experiments: The paper is a great theory paper, and the authors clearly demonstrate compatibility of their results with published experiments. The authors also show a few new experimental results on cell growth and fitness under various conditions. However, there seems to be no direct new experiment that confirms the simulations (e.g. fluorescent imaging of the molecules involved in the authors model). To me, this is perfectly fine, as the paper is already very strong with the new model as presented. However, I suggest the authors rephrase sentences such as ‘Our theory, confirmed by experimental analysis, reveals...’ to ‘Our theory, which is compatible with published experiments ...’. As the authors indicate themselves, many new experiments should be performed to validate the predictions (and therefore, fully validate the model).

We have rephrased the statement regarding the compatibility with experiments as suggested by the referee.

Title: I suggest a title that does not focus on adaptability or evolution. I really like Figure 5 and the text in the Conclusions and Discussions section, suggesting an evolutionary mechanism. However, I suppose the authors placed in into this section (and not into Results) exactly for the reason that this is a suggestion of how things could have evolved. There is no strong confirmation for this yet. Hence, I would choose a title that is more representative of the results for which the authors obtain strong support, e.g. robustness and the different submodules, rather than adaptability and evolution.

We thank the referee for his positive assessment of our hypothesis on Bem1 evolution as shown in Fig. 5. We agree that our title might have not represented the content of the paper well and have therefore chosen a new title: “Redundancy and the role of protein copy numbers in the cell polarization machinery of budding yeast.”

Language: Is protein dosage the best word? I would simply say: protein copy numbers, as this is apparently what the authors refer to?

We have changed the terms “expression level” and “protein dosage” to “protein copy number” throughout the manuscript.

Language: Occasionally sentences are very long and hard to digest. Example: ‘This significant difference in Cdc42 expression level sensitivity between the WT mechanism and the rescue mechanism is in the context of our theory explained by the

qualitative difference of their principles of operation, as we discussed above in the The Cdc42 interaction network facilitates a latent polarization mechanism'. 'Suggestion: break such long sentences down into 2-3 parts.

We thank the referee for bringing this issue to our attention. We have checked the manuscript for particularly long sentences and broken them up to improve readability.

L5: the polarity machine is robust to genetic perturbations: is this statement true in generality? I am not a yeast expert, could the authors make sure that this sentence is precise? E.g., is there robustness with respect to the deletion of any (known) individual gene of the polarity machinery? PS: after reading the whole paper it seems very misleading – the authors talk about robustness with respect to a single gene.

Our findings show that the Cdc42 polarization machinery exhibits a high degree of redundancy on the level of functional submodules. Knocking out any one of the three these submodules leaves the polarization machinery intact. Since each submodule comprises multiple proteins, this means that there is robustness against many protein deletions/mutations. This includes mutations of Cdc42 (binding it strongly to the membrane by fusion with Snc2 or psy1TM). However, the polarization machinery is not robust against the deletion of Cdc42, which is the central polarity organizer.

In the discussion, we focus on one protein specifically, Bem1, because it has long been claimed to be essential for Cdc42 polarization and because it was the subject of a previous evolution-in-the-lab experiment.

We have added a sentence clarifying this switch from generic to specific in the discussion.

L35/L426: 'Active Cdc42 directs both cytosolic diffusion: Diffusion itself is obviously random movement and not 'directed'. Of course, as the authors explain, they mean diffusion driven by concentration gradients. However, diffusion inevitably requires a gradient; hence I think the word 'directed' is misleading in the present context. I suggest removing 'directed' when talking about diffusion, and just using 'directed' in the context of transport along actin cables.

We deliberately use the term directed transport both for actin-based transport and diffusive flux driven by a sustained concentration gradient, to emphasize that both play interchangeable roles in the spatial self-organization of proteins. To avoid confusion, we rephrased slightly to highlight that a concentration must be maintained in order for a directed diffusive flux to occur.

L43.. Clearly state that Cdc24 is the GEF

Done

L166: 'As expected, WT cells grow at all galactose concentrations.' . If I understand it correctly, Fig. 3 is normalised to wild type. Hence, by definition, WT has a fitness of 1. So how can we see that this statement is true? Ok, if the fitness is rescaled to 1, we

know it is not zero, but it could be very small. It may be good to show the not-rescaled values somewhere in the SI (Table S6 seems to show only the fraction of wells with proliferation. Or is that how the authors define the fitness? Is a precise definition of fitness given anywhere?)

We have now included the non-normalized fitness in Fig. S5. Because of the varying galactose concentrations, there are osmolarity differences across media. This will also influence the WT fitness, which is seen by a trend for this background as well. In order to make the trend of our interest, namely the effect of varying Cdc42 on fitness, more clear, we therefore normalized all fitness values to the WT fitness value in each medium condition.

We defined fitness and the growth rate relative to WT.

L56 typo: 'a' missing

L159, 186 and maybe elsewhere: reference was broken. The authors should check thoroughly the references before the resubmission

L353: typo: =

We thank the referee for pointing us to these typos which we have fixed.

SI p3: The authors cite SI reference [22] for their argument that dissociation of the Cdc42 complex with a GAP is rate-limiting. First, do they mean rate-limiting compared to any other rates in the model (e.g. transport rates)? Second, if the dissociation is slow, is a Michaelis-Menten approximation justified, which was used in [22]? Also, [22] seems to deal with human proteins in vitro – are these results transferable to yeast (the main text line 114 seems to indicate the reference is about yeast, but glancing at the paper it does not appear to be so). Finally, the determined parameter k_{gt} in Table S5 seems to be of the same order of magnitude as the other rates (or even larger in the scenario for Fig. 4)

By rate limiting, we mean that the catalysis (and dissociation) step is slow compared to the binding step in the enzyme kinetics. We have clarified this in the SI. This is exactly the regime where the Michaelis-Menten approximation is justified. Moreover, it implies that a significant fraction of GAPs is bound in Cdc42-GAP complexes when the number of Cdc42 molecules exceeds the number of GAPs.

Indeed, [22] deals with human proteins in vitro. We have clarified this in the SI. Unfortunately we did not find a reference on the enzyme kinetics of Cdc42-GAPs in yeast. However, since Michaelis-Menten kinetics is a common assumption for enzymes, we believe that the result from human Cdc42 is transferable.

SI p3: The authors argue that Cdc24 is distributed uniformly on the membrane. I do not fully understand the argument. If Cdc24 can interact with Cdc42 which is non-uniform on the membrane, then Cdc24 should also possibly become non-uniform on the membrane due to these interactions. Lower GEF activity does not seem

necessary to imply this. It may depend on what exactly lower GEF activity means. If the binding of Cdc24 to 42 is the same but the subsequent rate of Cdc42 activation (e.g. GTP binding) is lower, I am not sure if that would imply that there is less of a non-uniformity of Cdc24 than with full GEF activity. Moreover, a 50% reduction in activity is not that high. It would be better if the authors could quickly check their assumptions, which should not be too hard.

The referee raises two related questions here. The conditions under which the concentration of Cdc24 remains uniform on the membrane and the role of reduced GEF activity.

We make the simplifying assumption that there is no long lived complex of Cdc24 and Cdc42 and that Cdc42 only interacts with the Cdc24 on the membrane, i.e. that it is not recruited from the cytosol. If one would include such processes, there could indeed be a non-uniform concentration of Cdc24 due to the interaction with Cdc42 and this might add further to the robustness of pattern formation. However, we want to emphasize that polarization is possible even if the GEF is distributed uniformly.

Regarding the GEF activity: Our point here is that increasing the GEF activity facilitates polarization for larger GAP concentrations, again independently of non-uniform GEF concentration.

A more detailed modeling of the interaction between Cdc42 and its GEF is certainly interesting. However, such model extensions are beyond the scope of the current work and would distract from the central topic of the manuscript: the redundancy of functional submodules in the Cdc42 polarization machinery.

Sl p6: The authors argue about the similarities between vesicle-based and diffusion-based transport. However, I would expect vesicle based transport along the cytoskeleton to lead to an advection instead of a diffusion term.

We agree that on a coarse-grained level, cytoskeletal transport can be described by an advection term which is directed by the local net polarization of cytoskeletal filaments (actin cables)

$$\partial_t c = \nabla \mathbf{J}_A, \quad \mathbf{J}_A = c \mathbf{p}$$

Importantly, diffusion can also be written in a closely related form, by explicitly writing out the diffusive flux in terms of the concentration gradient

$$\partial_t c = \nabla \mathbf{J}_D, \quad \mathbf{J}_D = -D \nabla c$$

Thus the gradient replaces the local cytoskeletal polarization. In the cell polarization machinery of budding yeast, the polarization of the cytoskeleton is established and maintained by a localized concentration of Cdc42-GTP (via formins). Similarly, protein recruitment to the membrane, which sets up and sustains cytosolic concentration

gradients, is downstream of Cdc42-GTP. Therefore, we can assume $\mathbf{p} \sim \nabla c$, such that cytoskeletal transport effectively is described by a diffusion equation.

We have added a short passage in the SI where we make the above argument.

SI p8/9: It appears there are 7 membrane bound species, yet the vector containing all membrane reactions, eq (5), has only 6 elements. Did I miss anything?

This was in fact a typo.

SI p15, step I. Typo 0:15x10⁻²?

Fixed.

SI p18, Step III: Why is $k_d = 0$ modelling the lack of recycling via the cytosol? Could one not argue no recycling corresponds to the attachment rate being zero, as opposed to the detachment rate?

$k_d = 0$ models strong membrane binding of Cdc42. This implies lack of recycling where most proteins reside on the membrane. Conversely, in the case of the attachment rate being zero (i.e. weak protein attachment) there would also be no recycling but most proteins would reside in the cytosol.

SI p18, bottom: 'that is unlikely' -> it missing; SI, p18: 'to same small value' -> the same

Done.

SI p22, bottom. Are the authors really referring to Video 6 here? Also, while it appears that the pattern is indeed stable after removal of the stimulus, have the authors explicitly shown this, at least through much longer simulations to confirm nothing changes?

Yes, the pattern is stable even for very long simulation times.

REVIEWERS' COMMENTS

Reviewer #1 (Remarks to the Author):

The authors have addressed my questions in the response letter. However, I would like to suggest that the authors include their clarifying response to point 4 to the corresponding relevant section of the paper, so that other readers do not have the same question. The full response to point 2 would also be helpful to have in the manuscript or SI. A partial response to this comment seems to be included on page 3 of the SI (where I note a typo "[FB:refsfromref.2]")

Reviewer #2 (Remarks to the Author):

The authors' response and revisions have satisfactorily addressed most of my major comments on the earlier version of the manuscript, therefore I support publication in its current form.

Reviewer #3 (Remarks to the Author):

The authors have either addressed my comments or defended their modelling choices. I can now recommend the manuscript for publication.

Typos identified:

335: on of one

498: it based on -> it is based